# Accessing parity-forbidden *d-d* transitions for photocatalytic $CO_2$ reduction driven by infrared light

Xiaodong Li [1,5], Li Li [2,5], Guangbo Chen [3], Xingyuan Chu[3], Xiaohui Liu[3], Chandrasekhar Naisa[3], Darius Pohl [4], Markus Löffler[4] & Xinliang Feng [1,3] ✉

A general approach to promote IR light-driven $CO_2$ reduction within ultrathin Cu-based hydrotalcite-like hydroxy salts is presented. Associated band structures and optical properties of the Cu-based materials are first predicted by theory. Subsequently, $Cu_4(SO_4)(OH)_6$ nanosheets were synthesized and are found to undergo cascaded electron transfer processes based on *d-d* orbital transitions under infrared light irradiation. The obtained samples exhibit excellent activity for IR light-driven $CO_2$ reduction, with a production rate of 21.95 and 4.11 µmol $g^{-1}$ $h^{-1}$ for CO and $CH_4$, respectively, surpassing most reported catalysts under the same reaction conditions. X-ray absorption spectroscopy and in situ Fourier-transform infrared spectroscopy are used to track the evolution of the catalytic sites and intermediates to understand the photocatalytic mechanism. Similar ultrathin catalysts are also investigated to explore the generality of the proposed electron transfer approach. Our findings illustrate that abundant transition metal complexes hold great promise for IR light-responsive photocatalysis.

In the pursuit of global carbon neutrality, artificial photocatalysis offers a powerful solution to simultaneously address the issues of the greenhouse effect and energy shortage. This process involves the direct transformation of carbon dioxide ($CO_2$) and water into useful fuels and oxygen under ambient conditions[1]. The proper photocatalysts with well-matched band edge positions are the prerequisite requirement to achieve this objective. In principle, the conduction band minimum (CBM) of the catalysts should be more negative than the potential for $CO_2$ reduction, while the valance band maximum (VBM) needs to be more positive than the potential for water oxidation. For this purpose, the band gaps of the photocatalysts should ideally fall within the range of 1.8–2.0 eV[2]. Recently, the infrared (IR) light-driven redox reactions have gained widespread attention among researchers. Despite the low energy of IR light and its tendency to generate localized heat, its relatively high proportion in the solar spectrum (ca. 50%) has prompted people to explore ways to utilize it. In principle, IR light is theoretically precluded for photocatalytic $CO_2$ reduction owing to its low photonic energy of less than 1.55 eV[3]. Fortunately, band structure engineering provides a viable strategy to fine-tune the energy band distribution and optimize the transfer behavior of activated electrons[4]. In this scenario, the step-by-step electron transition mode has proven to be the most attractive strategy to effectively utilize the low-energy photons, which could trigger the concurrent transformation of $CO_2$ and water into hydrocarbons and oxygen under IR light irradiation[5,6].

Recently, some ultrathin metallic catalysts (such as CuS and $CoS_2$ nanosheets) have been reported to offer an activity for IR light-driven $CO_2$ photoreduction, in which the intraband and interband electron transitions are coupled to realize the absorption of IR light while enabling the redox potential requirements of electron-hole pairs

[1]Max Planck Institute of Microstructure Physics, Weinberg 2, Halle 06120, Germany. [2]Hefei National Laboratory for Physical Sciences at Microscale, University of Science and Technology of China, Hefei, P. R. China. [3]Faculty of Chemistry and Food Chemistry & Center for Advancing Electronics Dresden (cfaed), Dresden University of Technology, Dresden 01062, Germany. [4]Dresden Center for Nanoanalysis (DCN), Dresden University of Technology, Helmholtzstreet, Dresden 01069, Germany. [5]These authors contributed equally: Xiaodong Li, Li Li. ✉e-mail: Xinliang.Feng@tu-dresden.de

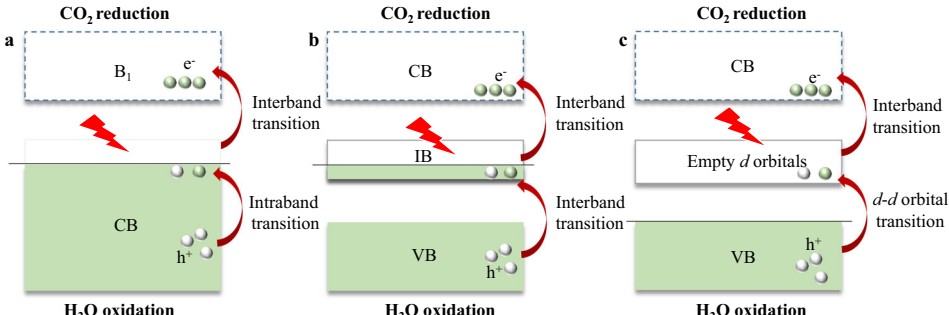

**Fig. 1 | Possible electron transition mechanisms for IR light-driven CO$_2$ reduction. a** Metallic catalysts (the energy bands below the Fermi level with fully occupied electrons are regarded as CB and the first band above the Fermi level without electron occupation is called to B$_1$). **b** Semiconductors with the intermediate band; **c** Hydrotalcite-like hydroxy salts with *d-d* orbital transition. Green ball: electrons; White ball: holes; Red lightning: IR light irradiation; Red arrows: electron transfer pathway; CB: conduction band; VB: valence band; B$_1$: Lowest empty band; IB Intermediate band; − Fermi level.

(Fig. 1a)[7,8]. Besides, some bismuth-based catalysts, like metallic Bi$_{19}$Br$_3$S$_{27}$ nanowires[9], has also been reported to exhibit excellent activity for CO$_2$ reduction under near-infrared (NIR) light irradiation. However, metallic conductors usually suffer from severe recombination of the photoexcited charge carriers, resulting in low efficiency for CO$_2$ photoreduction. Meanwhile, the serious electron scattering effect and localized plasmon effect of the metallic conductor can give rise to an energy loss of the active electrons and localized lattice heating, leading to the deactivation of the catalysts[10,11]. On the other hand, the defect-induced semiconductors with intermediate bands (e.g., WO$_3$ nanosheets with oxygen vacancies) also exhibit certain performance for IR light-driven CO$_2$ reduction, in which the reasonably regulated intermediate band could serve as a step for the cascaded electron transfer processes under IR light excitation (Fig. 1b)[3]. Nevertheless, the limited materials selection and harsh processes of defect regulation retard their further development. As such, there is an urgent need to investigate photocatalytic systems that follow a universal principle for the transition of charge carriers under IR light. Such systems would expand the range of photocatalysis to the IR spectrum and enable the preparation of full-spectrum-response catalysts by combining infrared photoactive catalysts with conventional catalysts.

Transition metal ion complexes are diverse and can be easily prepared, usually exhibiting a high IR light absorption capacity due to the *d-d* interband transitions[12,13]. In general, the transitions of electrons in *d* orbitals are parity-forbidden because of the same angular momentum quantum number. However, in some transition metal complexes with tetrahedral or octahedral coordination, like hydrotalcite-like hydroxy salts, the strong *p-d* orbital coupling between the coordinating groups and the metal ion can result in the degeneracy of valence band *d* orbitals of transition metals[14]. Therefore, some empty *d* orbitals appear in the middle of the band gap, which could act as a "cushion step" to make the *d-d* transition possible and thus induce a strong IR light absorption (Fig. 1c). Nevertheless, like most bulk materials, the excited carriers under IR light irradiation recombine rapidly and cannot be utilized for the catalytic reaction. On account that two-dimension (2D) materials possess ultrathin thickness in one dimension, they could have great potential for electron-hole separation benefiting from the shorter distance for carrier transfer[15–17]. To this end, the hydrotalcite-like hydroxy salts with 2D nanosheet structure could be a good candidate for IR light absorption and charge carrier separation so that the activated electrons participate in the photocatalytic reactions.

In this work, we first predict the existence of splitting *d* orbitals in 2D hydrotalcite-like hydroxy salts and their potential IR response-ability by the simulated density of states (DOS), band structures, and optical absorption spectra. Then, ultrathin basic copper sulfate (Cu$_4$(SO$_4$)(OH)$_6$) nanosheets (CSON) were prepared as the prototype, and an IR light-driven CO$_2$ reduction reaction was taken as an example to verify the possibility of *d-d* transition-induced photocatalytic reactions. The basic OH groups on the surface of Cu$_4$(SO$_4$)(OH)$_6$ nanosheets provide abundant sites for initial CO$_2$ adsorption. Meanwhile, the low-coordination tetrahedral Cu in the CSON is confirmed by X-ray absorption near edge structure (XANES) analyses and Auger electron spectroscopy, which could act as the active site for CO$_2$ activation and continuous protonation. Ultraviolet-visible-near-infrared (UV-vis-NIR) spectra and Gibbs free energy diagrams demonstrate that the tetrahedral Cu sites could not only improve the IR light absorption but also reduce the reaction energy barrier for CO$_2$ reduction. Combining Tauc plot, density functional theory (DFT) calculations, and in situ FTIR spectrum, we propose a photocatalytic mechanism of the activated parity-forbidden electron transfer of *d-d* orbitals under IR light irradiation (Fig. 1c). As a result, the ultrathin CSON exhibit effective production of CO and CH$_4$ with the evolution rate of 21.95 and 4.11 µmol g$^{-1}$ h$^{-1}$, respectively, under IR light irradiation, outperforming most reported catalysts under the same reaction conditions (yield < 20 µmol g$^{-1}$ h$^{-1}$). We further demonstrate the universality of IR light-driven CO$_2$ reduction based on the *d-d* electron transfer mechanism by preparing and evaluating various 2D hydrotalcite-like hydroxy salts, like Cu$_2$(NO$_3$)(OH)$_3$ nanosheets (CNON), Cu$_3$(PO$_4$)(OH)$_3$ nanosheets (CPON) and Cu$_2$(CO$_3$)(OH)$_2$ nanosheets (CCON), which provide robust catalyst systems with high performance for photocatalytic CO$_2$ reduction.

## Results
### Theoretical design and simulations of 2D hydrotalcite-like hydroxy salts
We first construct the model of 2D CSON with the thickness of a single-unit cell for theoretical simulations (Fig. 2a). [100] plane is selected as the exposed surface since it is predicted to have the highest surface energy of 1.03 J m$^{-2}$ compared with [010] facet (0.78 J m$^{-2}$) and [001] facet (0.67 J m$^{-2}$) (Supplementary Fig. 1), suggesting [100] plane has the most excellent activity for catalytic reactions. Moreover, the calculated optical absorption spectra (Fig. 2b) show that there are significant absorption edges in the IR light region (0.6−1.6 eV), which could be caused by the *d-d* electron transfer. And the intrinsic band gap for photon absorption is around 3.0 eV, indicating that the redox potential of CSON is large enough for simultaneously catalyzing CO$_2$ reduction and water oxidation.

The DOS and band structure are further calculated to unveil the distribution of orbitals and possible transfer pathways for the excited electrons. As expected, there are apparent splitting *d* orbitals in the band gap of the spin-down DOS of CSON (Fig. 2c), which is fully consistent with the corresponding band structure shown in Fig. 2d. According to the projected density of state (PDOS) (Fig. 2c), the occurred empty *d* bands located in the band gap are attributed to the

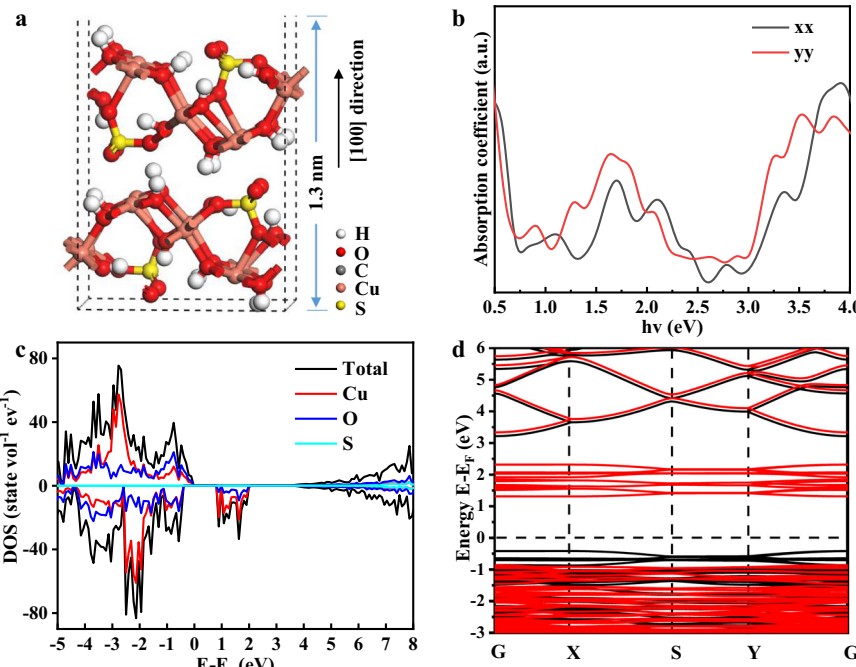

**Fig. 2 | Theoretical simulations of ultrathin 2D CSON. a** Theoretical slab model for CSON with the thickness of 1.3 nm along [100] direction. **b** The simulated optical absorption spectra, (**c**) DOS and (**d**) energy band structure of ultrathin 2D CSON slab. Black and red lines in (**b**) mean the optical absorption along x and y directions of the slab. Black and red lines in (**d**) represent the spin up and spin down energy band respectively. Source data are provided as a Source Data file.

degeneracy of *d* orbitals in Cu atoms. The hybridization of *d* orbitals in transition metal atoms and *p* orbitals in OH ligands leads to the splitting of the metal *d* electron orbitals, shifting part of the Cu *d* bands level up to form the empty *d* bands, which could act as a "cushion step" for *d-d* electron transition and achieve the cascaded electron transition process. In this respect, valence electrons absorb the IR-light photons and get excited to the middle empty *d* bands. Subsequently, these activated metastable electrons continue to absorb IR light and transition to the conduction band, enabling them to participate in the $CO_2$ reduction reaction. Meanwhile, the remaining valence band holes can be utilized for the $H_2O$ oxidation reaction (Fig. 1c). Therefore, the *d-d* electron transition mechanism and suitable bandwidth of 2D CSON provide a high potential for IR light absorption and possible activity for IR light-driven $CO_2$ photoreduction. Similarly, theoretical slab models of CNON, CPON and CCON are also analyzed by DFT calculations. According to the obtained DOS and band structures (Supplementary Fig. 2), the splitting empty *d* orbitals exist in all the Cu-based hydrotalcite-like hydroxy salts (CHHS), indicating the universality of *d-d* electron transfer mechanism and the huge potential of 2D CHHS for IR light-driven photocatalysis.

## Synthesis and characterizations of 2D CSON

Guided by the theoretical simulations, we then synthesized the ultrathin 2D CSON through the solvothermal method (see details in Methods). TEM and HRTEM images in Supplementary Fig. 3a, b show that the obtained CSON have a flake-like morphology with the exposed plane along the [100] direction. The corresponding energy-dispersive spectroscopy (EDS) element mapping images verify the homogeneous distribution of Cu, S and O elements (Supplementary Fig. 3c–f). The thickness of 2D CSON is determined to be ~1.3 nm by atomic force microscopy (AFM) (Supplementary Fig. 4), which is consistent with the theoretical thickness of its single-unit cell. Compared with its bulk counterpart, CSON exhibit a high ratio of surface Cu sites with tetrahedral coordination in theory (Supplementary Fig. 5).

Previous studies have suggested that the tetrahedral Cu atoms generally exhibit higher extinction coefficients in the IR light region,

which are expected to be around 50–100 times larger than that of octahedral Cu[13]. In this regard, we also prepared the calcined $Cu_4(SO_4)$ $(OH)_6$ nanosheets (c-CSON) with abundant surface vacancies by calcining the pristine $Cu_4(SO_4)(OH)_6$ nanosheets (p-CSON) at 300 °C in air (see details in Methods), in which the surface $-SO_4$ vacancies will induce the generation of more tetrahedral Cu atoms. Thermogravimetry-differential scanning calorimetry (TG-DSC) and thermogravimetry-differential thermal gravity (TG-DTG) curves were conducted to explore the structural stability of CSON during the calcination process (Supplementary Fig. 6). The similar TG-DSC and TG-DTG curves of p-CSON and c-CSON confirm that the crystal phase and composition of the two samples are almost identical before and after calcination. Meanwhile, the DSC curves of both p-CSON and c-CSON show that there is no recrystallization or melting process before 300 °C, and only a recrystallization peak occurs at 552–560 °C, followed by a melting peak at 717–718 °C. TEM images in Fig. 3a directly show the clear microporous structure on the surface of c-CSON. Compared with the p-CSON, there is no obvious change in the whole 2D morphology, crystallinity, and element distributions, as displayed in Fig. 3b, c. AFM images in Fig. 3d and Supplementary Fig. 7 confirm that the c-CSON maintain the ultrathin but a porous surface with uneven height profiles. Besides, the X-ray diffraction (XRD) pattern, Fourier-transform infrared spectroscopy (FTIR) spectrum (Supplementary Fig. 8) and X-ray photoelectron spectroscopy (XPS) analysis (Supplementary Fig. 9) illustrate that the phase and structure of 2D CSON after calcination are well retained. According to the EDS (Supplementary Fig. 10) and XPS composition analyses (Supplementary Table 1), the c-CSON exhibit reduced S content compared with p-CSON, which further corroborates the existence of surface $-SO_4$ vacancies.

To further demonstrate the tetrahedral-coordination Cu sites, X-ray absorption near-edge structure (XANES) analyses were performed to determine the valence and coordination environment. As depicted in Fig. 3e and the insert, compared with p-CSON, the Cu K-edge position of the c-CSON shifts to a lower energy and the intensity of the white line at round 8996 eV for c-CSON is much weaker. These verify a lower coordination of Cu atoms in c-CSON[18,19]. The

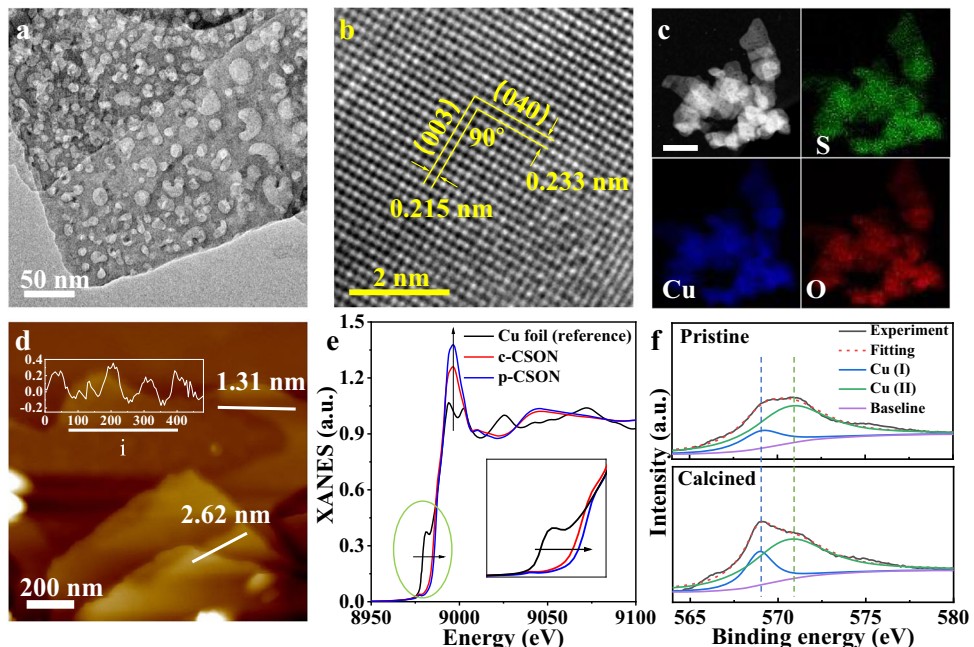

**Fig. 3 | Characterizations of c-CSON. a** TEM image. **b** HRTEM image, in which the exposed facet can be inferred along [100] direction because 0.215 nm and 0.233 nm interplanar distances match well with the $d_{003}$ and $d_{040}$ spacings, and the corresponding dihedral angle of 90° agrees well with the calculated angle between the (003) and (040) planes. **c** Annular dark-field TEM images and corresponding elemental mapping images, scale bar: 200 nm; (**d**) AFM images, in which the obtained nanosheets is confirmed to be ca. 1.31 nm with a hole-rich surface. Insert: the corresponding height profiles of i area (unit: nm). **e** Cu K-edge XANES spectra, the insert is a zoomed-in view of the part of the green circle. **f** Cu LMM Auger electron spectroscopy in p-CSON and c-CSON, in which the ratio of Cu (I) component of the calcined samples is remarkably increased after calcination. Source data are provided as a Source Data file.

Fourier-transformed extended X-ray absorption fine structure (FT-EXAFS) profiles (Supplementary Fig. 11) show a main peak at about 1.89 Å in both c-CSON and p-CSON, which could be attributed to the Cu-O scattering path, fairly agreeing with the Cu-O bonds (1.91 Å) in theoretical models. The Cu LMM Auger electron spectroscopy was further conducted to reveal the Cu coordination information in p-CSON and c-CSON. As shown in Fig. 3f, the ratio of low-coordination Cu atoms in c-CSON is significantly increased compared to that in the pristine one, indicating the increased tetrahedral Cu sites after the calcination. Moreover, the calculated optical absorption spectra (Supplementary Fig. 12a) confirm that c-CSON also exhibit IR light absorption edges between 0.6 and 1.6 eV. The corresponding DOS and band structure of c-CSON (Supplementary Fig. 12b, c) show that there are some similar splitting $d$ orbitals in the band gap as in p-CSON. Based on the DFT predictions and experimental characterizations presented above, it is evident that both the synthesized p-CSON and c-CSON have the ability to absorb IR light through the $d$-$d$ electron transition mechanism. This makes it possible to drive $CO_2$ reduction using IR light.

### *d-d* orbital transition processes and IR light-driven $CO_2$ reduction performance

UV-vis-NIR diffuse reflectance spectra were conducted to evaluate the IR light response of p-CSON and c-CSON (Fig. 4a). The strong absorption of IR light by both p-CSON and c-CSON is evident, with the calcined sample exhibiting a higher absorption intensity. This observation is consistent with the color change of 2D CSON from light blue to dark green after calcination (inset of Fig. 4a). The corresponding Tauc plots in Fig. 4b, c illustrate that there are two obvious IR absorption edges around 1.09 and 1.17 eV in p-CSON, and 1.07 and 1.18 eV in c-CSON, fairly agreeing with the theoretical optical absorption spectra. Such IR light absorption is attributed to the electron transfer from the valence band to the splitting empty $d$ band and subsequent continuous transfer to the conduction band, indirectly

confirming the $d$-$d$ transition and cascaded electron transfer processes. To reveal the precise band structure, ultraviolet photoelectron spectroscopy (UPS) was further conducted. As displayed in Fig. 4d–f and Supplementary Figs. 13, 14, for electrode potential (V vs. NHE) at pH = 0, the positions of the CBM for p-CSON (−0.47 V) and c-CSON (−0.31 V) are more negative than the potential of $CO_2$ reduction to CO (−0.11 V) and $CH_4$ (0.16 V), while the VBM for p-CSON (2.70 V) and c-CSON (2.42 V) are more positive than the potential for $H_2O$ oxidation to $O_2$ (1.23 V). These results are well-consistent with those obtained from Mott-Schottky plots presented in Supplementary Fig. 15, clearly suggesting the ability for $CO_2$ conversion and oxygen generation over p-CSON and c-CSON.

To evaluate the catalytic activity of p-CSON and c-CSON for IR light-driven $CO_2$ photoreduction, photocatalytic experiments were performed under IR light irradiation (simulated by the solar simulator with the AM1.5 and CUT800 filters) without any sacrificial agent at ambient temperature (see more details in the Methods section). The corresponding spectrum of our solar simulator is displayed in Supplementary Fig. 16, in which the IR light component is obtained by using the CUT800 filters. Since the IR light tends to generate heat and raise the temperature of the photocatalyst and reactor[20,21], a water-cooled gas-solid reaction system (Supplementary Fig. 17a, b) was designed and applied in this work to avoid the effects of thermo-catalysis. The average temperature of CSON-based thin film during IR light irradiation has proved to be almost unchanged by the in situ thermographic photographs (Supplementary Fig. 17c–h). The performance evaluations in Fig. 4g and Supplementary Fig. 18 demonstrate that both p-CSON and c-CSON exhibit excellent activity for CO and $CH_4$ production under IR light irradiation, in which the calcined sample shows better performance with the CO and $CH_4$ evolution rate of 21.95 and 4.11 $\mu$mol $g^{-1}$ $h^{-1}$ respectively, outperforming most reported catalytic systems (yield < 20 $\mu$mol $g^{-1}$ $h^{-1}$) under the similar conditions (Supplementary Table 2). The product selectivity of CO and $CH_4$ is calculated to 84.2% and 15.8% for c-CSON, 85.7% and 14.3% for p-CSON;

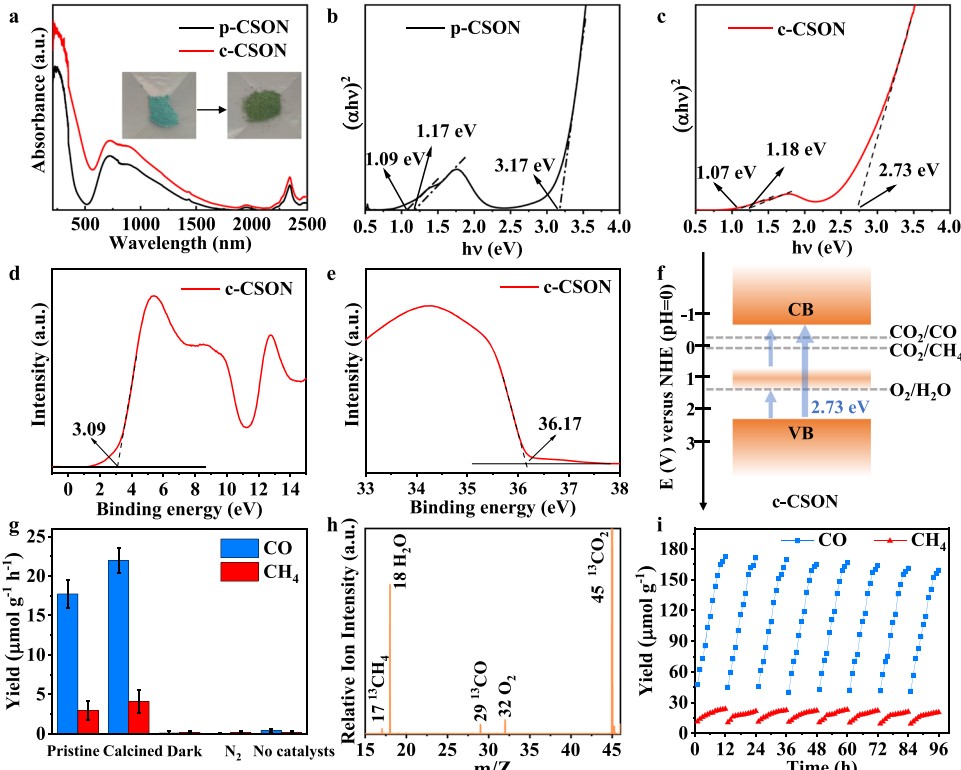

**Fig. 4 | Band structures of ultrathin 2D CSON and their performance for IR light-driven $CO_2$ reduction. a** UV-vis-NIR diffuse reflectance spectra; Tauc plots for (**b**) p-CSON and (**c**) c-CSON. **d, e** SRPES valence-band and secondary electron cutoff spectra of c-CSON, in which VBM is located at 3.09 eV below the Fermi level determined from Au (called as $E_{edge}$) and the energy of secondary electron cutoff is 36.17 eV. **f** Schematics illustrating the electronic band structures of c-CSON; blue arrows represent the electron transition process. CB, conduction band; VB, valence band. **g** Yields of photocatalytic $CO_2$ reduction to CO and $CH_4$ over different catalysts and conditions, error bars represent the standard deviation (s. d.) of three independent measurements using 5 mg fresh sample for each measurement. **h** SVUV-PIMS spectrum of the products after $^{13}CO_2$ photoreduction for c-CSON at $hv$ = 14.5 eV. **i** Cycling measurements for $CO_2$ photoreduction to $CH_4$ and CO on c-CSON (using 5 mg sample for this measurement; when a new catalytic cycle begins, the reactor is pumped and refilled with pure $CO_2$). Source data are provided as a Source Data file.

while the electron selectivity is 57.2 % and 42.8% for c-CSON, 59.9 % and 40.1 % for p-CSON. Moreover, the apparent quantum yield of c-CSON for IR light-driven $CO_2$ reduction (Supplementary Fig. 18b) is calculated up to 0.122% under the monochromatic light of 800 nm wavelength, which is quite high compared with the previously reported IR-light catalysts, such as $WO_3$ and $CuS^{3,7}$. As for the oxidation end, the remaining holes in the valence band are consumed by water to generate $O_2$ (Supplementary Fig. 19). No other liquid or gas products are detected by gas chromatography (GC) (Supplementary Fig. 20) and nuclear magnetic resonance (NMR) spectrum (Supplementary Fig. 21).

In addition, several controlled experiments were conducted to explore the IR light-driven $CO_2$ reduction reaction over the CSON, including experiments conducted in the dark, without $CO_2$, and without catalysts. As shown in Fig. 4g, these experiments confirm that IR light, $CO_2$ reactant, and catalysts are all necessary for IR light-driven $CO_2$ reduction. The isotope-labeled $^{13}CO_2$ mass spectrometry for c-CSON (Fig. 4h and Supplementary Fig. 22a, b) and p-CSON (Supplementary Fig. 22c) was performed to further unveil the source of CO and $CH_4$ production, in which the photon energy of 14.5 eV is selected for distinguishing the CO and $CH_4$ according to the absolute photoionization cross sections for $CO_2$, CO and $CH_4$ in Supplementary Fig. 23. As a result, only $^{13}CO$ and $^{13}CH_4$ species are detected after using the isotope-labeled $^{13}CO_2$ as reactants, implying that the products indeed originate from IR light-driven $CO_2$ reduction for both of them. It is worth noting that c-CSON do not show any significant performance degradation after 96 h long-term photocatalysis (Fig. 4i). The excellent stability of the catalysts is further confirmed by the TEM, HRTEM, XRD patterns, and FTIR spectra as the morphology and crystal

structure for c-CSON remain unchanged before and after 96 h continuous photocatalysis (Supplementary Fig. 24).

## Mechanistic insights into the $CO_2$ photoreduction processes

To reveal the electron transition processes by the photoexcitation, the corresponding carrier dynamics characterizations were performed as shown in Supplementary Fig. 25. The transient photocurrent response spectra (Supplementary Fig. 25a) demonstrate that p-CSON and c-CSON exhibit significant photocurrent density under IR light irradiation, indicating that both samples are capable of achieving IR light-driven excitation of electron-hole pairs and charge carrier separation. It is notable that c-CSON show higher photocurrent density and smaller transfer resistance (Supplementary Fig. 25a, b), illustrating the associated higher photoexcited carrier density and rapid electron transition processes. On the other hand, the longer fluorescence lifetime and lower intensity of photoluminescence (PL) spectroscopy indicate that the charge carriers in c-CSON could be easier to separate and participate in the catalytic reactions instead of recombination (Supplementary Fig. 25c, d). The excellent carrier dynamics observed in c-CSON can mainly be attributed to the presence of more surface Cu sites with tetrahedral coordination and the ultrathin 2D porous configuration. These characteristics not only enhance IR light absorption but also reduce the transfer distance, thereby accelerating charge carrier separation. This, in turn, is beneficial for subsequent IR light-driven $CO_2$ reduction.

To further understand the reaction mechanism, in situ FTIR spectroscopy was performed to identify the reaction intermediates during IR light-driven $CO_2$ reduction in real-time. As displayed in Fig. 5a

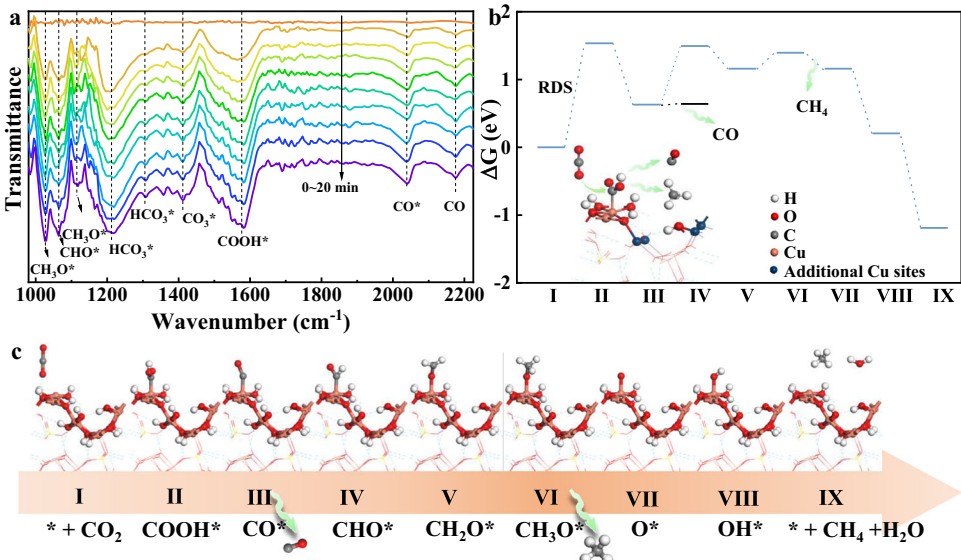

**Fig. 5 | Mechanistic studies of ultrathin 2D CSON for IR light-driven $CO_2$ reduction. a** In situ FTIR spectroscopy characterization for co-adsorption of a mixture of $CO_2$ and $H_2O$ vapor under light irradiation over c-CSON. **b** Gibbs free energy diagrams and (**c**) intermediate structures of c-CSON during $CO_2$ reduction under IR light irradiation. RDS: rate-determining step. Source data are provided as a Source Data file.

for c-CSON and Supplementary Fig. 26a for p-CSON, a series of new infrared peaks at around $1570\ cm^{-1}$ were detected for both samples, which are attributed to the COOH* group, a crucial intermediate for $CO_2$ reduction to CO or $CH_4$[22,23]. At the same time, the absorption bands near 1116 and $1028\ cm^{-1}$ belong to the $CH_3O^*$ group, and the peaks at $1064\ cm^{-1}$ are assigned to the characteristic bands of CHO*; both the $CH_3O^*$ and CHO* groups are pivotal intermediates of $CO_2$ photoreduction to $CH_4$[24–27]. Moreover, the peaks at 2042 and $2175\ cm^{-1}$ could be attributed to the adsorbed and free CO products respectively[28–30]. It is notable that the intensities of all these peaks for reaction intermediates and products gradually increased with extension of the irradiation time, suggesting the continuous photocatalytic reaction progresses. Besides, the peaks at 1305 and $1208\ cm^{-1}$ are inferred to the symmetric and asymmetric stretching of $HCO_3^*$ group, while the peaks at $1411\ cm^{-1}$ are indexed to the $CO_3^*$ group[22,31]. Based on the analysis of in situ FTIR, the most likely reaction pathway for this IR light-driven $CO_2$ reduction could be proposed as follows:

$$* + CO_2 + e^- + H^+ \rightarrow COOH^* \tag{1}$$

$$COOH^* + e^- + H^+ \rightarrow CO^* + H_2O \tag{2}$$

$$CO^* + e^- + H^+ \rightarrow CHO^* \text{ or } CO^* \rightarrow CO \uparrow + * \tag{3}$$

$$CHO^* + e^- + H^+ \rightarrow CH_2O^* \tag{4}$$

$$CH_2O^* + e^- + H^+ \rightarrow CH_3O^* \tag{5}$$

$$CH_3O^* + e^- + H^+ \rightarrow CH_4 \uparrow + O^* \tag{6}$$

$$O^* + e^- + H^+ \rightarrow OH^* \tag{7}$$

$$OH^* + e^- + H^+ \rightarrow H_2O + * \tag{8}$$

where the asterisks * represent the possible active sites in the catalysts[32].

Transition metals are noteworthy for their half-filled $d$ orbitals and high charge density, which allow them to not only provide additional orbitals for bonding with reactants but also participate in catalytic reactions by providing electrons[33]. In particular, Cu site has been extensively studied as an active center for $CO_2$ reduction reactions[34]. In comparing p-CSON with defective c-CSON, we observed a significant increase in low-coordinated Cu (tetrahedral Cu) in c-CSON. As a result, the product yield also increased, but the product selectivity remained largely unchanged. This suggests that c-CSON and p-CSON have similar active centers, with differences only in the amount of catalytic sites. Therefore, tetrahedral Cu is regarded as the active site for $CO_2$ reduction. According to the detected intermediates of in situ FTIR, the theoretical models of different reaction intermediates were constructed and the corresponding Gibbs free energy was also calculated to determine the reaction pathway and energy barrier of c-CSON (Fig. 5b, c) and p-CSON (Supplementary Fig. 26b, c) during IR light-driven $CO_2$ reduction. The energy of each elementary reaction is provided in Supplementary Tables 3–5. As the first step, $CO_2$ adsorption plays an important role in promoting the catalytic performance[35,36]. Given that the basic groups, like -OH and $-NH_2$ ligands, are usually regarded as the efficient active site for $CO_2$ adsorption benefiting from the acid-base neutralization effect[37,38], we first calculated the $CO_2$ adsorption energy on OH and Cu sites in CSON. As shown in Supplementary Fig. 27, $CO_2$ molecules prefer to be adsorbed on the surface OH sites than Cu sites due to the lower adsorption energy of the former (−0.02 eV for OH sites and 0.08 eV for Cu sites). More importantly, compared with p-CSON $(14.4\ m^2\ g^{-1})$, c-CSON with abundant surface vacancies possess about 2-times higher specific surface area $(24.2\ m^2\ g^{-1})$ according to the Brunauer-Emmet-Teller (BET) plots in Supplementary Fig. 28a, which is beneficial for $CO_2$ adsorption, as further confirmed by the $CO_2$ adsorption isotherms in Supplementary Fig. 28b. Subsequently, the adsorbed $CO_2^*$ would move to the nearby Cu sites to be activated and protonated for the formation of COOH* intermediates, which is considered as the rate-determining step (RDS) according to the calculated Gibbs free energy in Fig. 5b.

Moreover, the calculated Bader charge (Supplementary Fig. 29) and charge density distribution (Supplementary Fig. 30) demonstrate that the surface Cu sites in c-CSON have higher electron density than that in the pristine sample, which is helpful for $CO_2$

activation and continuous protonation processes. We further calculated the adsorption energy and charge density difference (CDD) of COOH* intermediates on p-CSON and c-CSON. As shown in Supplementary Fig. 31, c-CSON exhibit stronger interaction with the key intermediate of COOH* radicals, in which the adsorption energy on c-CSON is −1.55 eV, much stronger than on p-CSON (−1.16 eV). The corresponding CDD shows that electron transfer between COOH* intermediate and c-CSON is more than that in p-CSON (Supplementary Fig. 32). This means that the surface Cu sites in c-CSON are more favorable for adsorbing COOH* intermediates and thus lower the reaction energy barrier. This result fairly agrees with the calculated Gibbs free energy - the energy barrier of the RDS reduces from 1.925 eV for p-CSON to 1.535 eV for c-CSON. As displayed in Supplementary Fig. 33, different surface Cu sites in c-CSON show similar adsorption energy for COOH* intermediates, which suggests that the surface -$SO_4$ vacancies can significantly reduce the steric hindrance and provide more active Cu sites for $CO_2$ conversion, therefore promoting the performance of IR light-driven $CO_2$ reduction.

### The generality of *d-d* electron transfer approach for IR light-driven $CO_2$ reduction

We demonstrate that the *d-d* electron transition processes and the underlying reaction mechanisms for IR light-driven $CO_2$ reduction can also be extended to various hydrotalcite-like hydroxy salts. Therein, we prepared CNON, CPON and CCON samples. TEM and HRTEM images confirm their flake-like 2D thin sheet structures (Supplementary Fig. 34a, b, Supplementary Fig. 37a, b, and Supplementary Fig. 40a, b). The corresponding elemental mapping images illustrate the uniform distribution of each element in the counterpart nanosheets (Supplementary Fig. 34c–f, Supplementary Fig. 37c–f, and Supplementary Fig. 40c–f). According to XRD patterns, their phase is indexed to standard PDF card No. 75-1779 for CNON (Supplementary Fig. 35b), No. 72-0258 for CPON (Supplementary Fig. 38b) and No. 72-0075 for CCON (Supplementary Fig. 41b), respectively. Theoretical slab models show that all these nanosheets possess octahedral (in bulk) and tetrahedral (in surface) Cu coordination structures (Supplementary Fig. 35a, Supplementary Fig. 38a, and Supplementary Fig. 41a), indicating their potential IR light absorption abilities. As a result, UV-vis-NIR diffuse reflectance spectra (Supplementary Fig. 35c, Supplementary Fig. 38c, and Supplementary Fig. 41c) clearly demonstrate their similar optical absorption properties with CSON. More impressively, the occurred low-energy absorption edges located around 1.10 eV in the Tauc plots for CNON (Supplementary Fig. 35c), CPON (Supplementary Fig. 38c) and CCON (Supplementary Fig. 41c) are all attributed to the IR light-driven *d-d* electron transition, while the high-energy absorption edges (higher than 3 eV) represent the intrinsic electron transition from the valence band to the conduction band. To evaluate their catalytic activity for IR light-driven $CO_2$ photoreduction, photocatalysis experiments were performed under the same condition as that for CSON. As displayed in Supplementary Figs. 36, 39, and 42, the obtained CHHS delivered pronounced catalytic activity for IR light-driven $CO_2$ photoreduction, with the CO and $CH_4$ yields of 5.49 and 15.34 μmol g$^{-1}$ h$^{-1}$ for CNON, 1.81 and 1.03 μmol g$^{-1}$ h$^{-1}$ for CPON, 2.11 and 0.43 μmol g$^{-1}$ h$^{-1}$ for CCON, respectively.

## Discussion

In summary, we have certified a cascaded *d-d* electron transition mode over the 2D hydrotalcite-like hydroxy salts, which can be used as the general strategy for IR light-driven $CO_2$ reduction. Various ultrathin Cu-based hydrotalcite-like hydroxy salts are firstly synthesized, for which theoretical and experimental analysis confirm their catalytic performance for IR light-driven $CO_2$ photoreduction based on *d-d* electron transition mode. In situ characterizations and DFT calculations unveil the evolutions of active sites, charge carriers, and reaction intermediates. As the result, c-CSON exhibit excellent performance for IR light-driven $CO_2$ reduction to CO and $CH_4$ with the rate of 21.95 and 4.11 μmol g$^{-1}$ h$^{-1}$ respectively, outperforming most reported IR-responsive catalysts. This work has thus illustrated that the 2D transition metal hydrotalcite-like hydroxy salts have tremendous potential for IR light-driven $CO_2$ reduction, providing an opportunity for the development of IR-light-responsive catalysts and paving the way for the design of full-spectrum catalytic systems.

## Methods

### Materials

$CuSO_4 \cdot 5H_2O$ (ACS reagent, ≥98%), NaOH (ACS reagent, ≥97.0%, pellets), $Cu(NO_3)_2 \cdot 3H_2O$ (puriss. p.a., 99–104% (RT)), $Na_2HPO_4$ (ACS reagent, ≥99.0%), $CuCl_2 \cdot 2H_2O$ (ACS reagent, ≥99.0%), $Na_2CO_3$ (powder, ≥99.5%, ACS reagent), Hexadecyltrimethylammonium bromide (CTAB) (≥98%), Ethanol (absolute, suitable for HPLC, ≥99.8%) and Ammonium hydroxide solution (puriss. p.a., reag. ISO, reag. Ph. Eur., ≥25% $NH_3$ basis) are all acquired from Sigma-Aldrich and were used without any further purification. Deionized (DI) water with resistivity of 18.2 MΩ.cm is obtained by the ultra-pure water system from Stakpure GmbH, Germany.

### Catalysts synthesis

**Synthesis of p-CSON.** A 360 mg portion of $CuSO_4 \cdot 5H_2O$ was added into the mixture of 8 mL absolute ethanol and 10 mL deionized (DI) water. After vigorous stirring for 30 min, 18 mg hexadecyltrimethylammonium bromide (CTAB) was added into the above solution followed by stirring for another 30 min. Subsequently, 18 mL ammonia solution with a 1% volume concentration and 0.4 mL 1.9 M NaOH solution was slowly dropped into the mixture system. Afterwards, the solution was transferred into a 50 mL Tefon-lined autoclave, sealed and heated at 80 °C for 30 min, and allowed to cool to room temperature naturally. The final product was collected by centrifuging the mixture, washed with ethanol and deionized (DI) water for several times until the organic residuals were completely removed, and then dried in vacuum oven at 60 °C overnight. The light blue powder was obtained for further usage.

**Synthesis of c-CSON.** In a typical procedure, the as-obtained ultrathin 2D p-CSON were calcined at 300 °C with a heating rate of 10 °C min$^{-1}$ for 1 h under the air atmosphere and then cooled to room temperature. The obtained black green powder was collected for further characterization.

**Synthesis of CNON.** 2 mmol (483.2 mg) $Cu(NO_3)_2 \cdot 3H_2O$ was added into 30 mL DI water. After vigorous stirring for 10 min, 30 mL NaOH solution with a concentration of 0.1 M was dropwise added into the above mixture followed by stirring for 10 min. Afterwards, the mixture solution was further ultrasonicated for 45 min. The final product was collected by centrifuging the mixture, washed with ethanol and deionized (DI) water for several times until the organic residuals were completely removed, and then dried in vacuum oven at 60 °C overnight. The light blue powder was obtained for further usage.

**Synthesis of CPON.** 2 mmol (483.2 mg) $Cu(NO_3)_2 \cdot 3H_2O$ and 190 mg $Na_2HPO_4$ was added into 30 mL DI water. After vigorous stirring for 10 min, 15 mL NaOH solution with a concentration of 0.1 M was dropwise added into the above mixture followed by stirring for 1 h. The final product was collected by centrifuging the mixture, washed with ethanol and deionized (DI) water for several times until the organic residuals were completely removed, and then dried in vacuum oven at 60 °C overnight. The light blue powder was obtained for further usage.

**Synthesis of CCON**. 0.9 mmol (153.5 mg) $CuCl_2 \cdot 2H_2O$ was added into 30 mL DI water (A solution). 0.1 M NaOH and 0.2 M $Na_2CO_3$ aqueous solution was mixed and named as B solution. Then B solution was slowly added into A solution until the pH was around 8.2. Afterwards, the solution was transferred into a 50 mL Tefon-lined autoclave, sealed and heated at 80 °C for 12 h, and allowed to cool to room temperature naturally. The final product was collected by centrifuging the mixture, washed with ethanol and deionized (DI) water for several times until the organic residuals were completely removed, and then dried in vacuum oven at 60 °C overnight. The light blue powder was obtained for further usage.

## Characterization

TEM and HRTEM images were performed with a JEOL Jem F-200C TEM with an acceleration voltage of 200 kV. XRD patterns were obtained from a Philips X'Pert Pro Super diffractometer with Cu Kα radiation ($\lambda = 1.54178$ Å). X-ray photoelectron spectroscopy (XPS) spectra were acquired on an ESCALAB MKII system with Al Kα ($hv = 1486.6$ eV) as the excitation source. The binding energies obtained in the XPS spectral analysis were corrected for specimen charging by referencing C 1 s to 284.8 eV. UV-vis-NIR diffuse reflectance spectra were measured on a Perkin Elmer Lambda 950 UV-vis-NIR spectrophotometer. In-situ FTIR spectra were obtained by using a Thermo Scientific Nicolet iS50. Synchrotron-radiation photoemission spectroscopy (SRPES) was executed at the National Synchrotron Radiation Laboratory (NSRL) in Hefei, China. AFM measurements was performed using a Veeco DI Nano-scope MultiMode V system. Ultraviolet photoelectron spectroscopy (UPS) was performed at the Catalysis and Surface Science Endstation at the BL11U beamline of the National Synchrotron Radiation Laboratory (NSRL). The workfunction (WF) was determined by the difference between the photon energy and the binding energy of the secondary cutoff edge. To be exact, $E_B = hv - (E_K + 4.3 - 5.0)$ and WF = $hv - (E_{cutoff} - E_F)$ ($E_B$, binding energy; $hv$, photon energy; $E_K$, kinetic energy; $E_{cutoff}$, secondary cutoff edge; $E_F$, Fermi level; photon energy of 40.0 eV and a sample bias of −5 V applied to observe the secondary electron cutoff). $CO_2$ adsorption isotherms measurements for all the synthetic samples were carried out using an automatic microporous physical and chemical gas adsorption analyser (ASAP 2020 M PLUS). Fluorescence emission decay spectra were recorded with a DeltaFlex-NL (HORIBA Scientific) spectrometer. BET isotherms were conducted by 3 Flex Multiport Surface Characterization Analyzer from Micromeritics.

## IR light-driven $CO_2$ reduction tests

Before performing the $CO_2$ photoreduction performance, we fabricated the sample into a thin film: the sample was dispersed in deionized water to gain a concentration of about 1 mg mL$^{-1}$, and then, through spin-dropping 5 mL of the above dispersion on a quartz glass, followed by heat treatment at 65 °C for 30 min, the catalysts thin film could be achieved. During the $CO_2$ photoreduction process, the catalysts thin film floated on 50 mL of water in a quartz glass vessel with the homothermal condensate water, which could enable the catalysts thin film to retain a constant temperature of $290 \pm 0.2$ K. A MC-PF-300C Xe lamp with AREF (full spectrum reflectance 200–2500 nm), AM 1.5 G filter and 800 nm cutoff filter was used to simulate infrared light, the corresponding illumination spectrum of which in comparison with sunlight is displayed in Supplementary Fig. 16. Note that the distance from the lamp to the sample was ~10 cm, and the irradiation area of sample is around 9.62 cm$^2$ with an output light density of ~71 mW cm$^{-2}$. The instrument was initially evacuated for three times, afterwards, pumped by high-purity $CO_2$ to reach an atmospheric pressure. The gas products were quantified by the Agilent GC-8860 gas chromatograph equipped with TDX-01 column, thermal conductivity detector (TCD) and flame ionization detector (FID) while ultrahigh-purity argon was used as a carrier gas (FID detector for carbon-based

products and TCD detector for $H_2$). The liquid products were quantified by nuclear magnetic resonance (NMR) (Bruker AVANCE AV III 400) spectroscopy, in which dimethyl sulfoxide (DMSO, Sigma, 99.99%) was used as the internal standard. A recirculating cooling water system was used to keep the photocatalytic system at $290 \pm 0.2$ K under light irradiation.

The product selectivity for $CO_2$ reduction to CO and $CH_4$ has been calculated using the following equation:

$$\text{Product selectivity of CO(\%)} = [n(CO)]/[n(CH_4) + n(CO)] \times 100\% \quad (9)$$

$$\text{Product selectivity of } CH_4(\%) = [n(CH_4)]/[n(CH_4) + n(CO)] \times 100\% \quad (10)$$

The electron selectivity for $CO_2$ reduction to CO and $CH_4$ (2e$^-$ for CO, 8e$^-$ for the formation of $CH_4$) has been calculated using the following equation:

$$\text{Electron selectivity of CO(\%)} = [2n(CO)]/[8n(CH_4) + 2n(CO)] \times 100\% \quad (11)$$

$$\text{Electron selectivity of } CH_4(\%) = [8n(CH_4)]/[8n(CH_4) + 2n(CO)] \times 100\% \quad (12)$$

where $n(CH_4)$ and $n(CO)$ are the amounts of produced $CH_4$ and CO.

## Apparent quantum efficiency measurement

The apparent quantum efficiency was defined by the ratio the effective electrons to the total input photon flux of different single wavelength[3]. The number of effective electrons was determined by CO and $CH_4$ yields of 12 h photoreaction under a monochromatic light wavelength at 800, 850, 940 and 1064 nm. And the total input photon flux of different single wavelength was calculated by the incident light intensity, which was determined using a Silicon-UV enhanced actinometer. The quantum yields for the photocatalytic $CO_2$ reduction reactions in our paper were determined using the following equation:

$$\Phi(\%) = \frac{\text{CO molecules} \times 2 + CH_4 \text{ molecules} \times 8}{\text{incident photons}} \times 100\% \quad (13)$$

## In situ FTIR spectra experiments

All FTIR spectra were recorded on Thermo Scientific Nicolet iS50. The spectra were displayed in transmission units and acquired with a resolution of 4 cm$^{-1}$, using 64 scans. The dome of the reaction cell had two KBr windows allowing IR transmission and a third window allowing transmission of irradiation introduced through a liquid light guide that connects to the same IR-light lamp. The catalyst powders were first added to the reaction cell and then trace amounts of water were sprayed on the surface of catalysts. After degassed in $N_2$ atmosphere for 20 min, the gas flow was switched to high-purity $CO_2$ until the adsorption is saturated, then the reaction cell was sealed. Next, the FTIR spectra were recorded as a function of time to investigate the dynamics of the reactant adsorption in the dark and desorption/conversion under irradiation.

## Photoelectrochemical measurements

Photocurrent and electrochemical impedance spectroscopy (EIS) were performed using an electrochemical workstation (CHI 660E, CH Instruments, Shanghai, China). Specifically, 10 mg of catalysts were dispersed in a mixture containing 950 μL ethanol and 50 μL Nafion solution, then ultrasonic treated to form homogenous catalyst ink.

Then the catalyst ink was dipped on a polished FTO glass and dried in air. The photocurrent measurements were conducted in a three-electrode cell system under irradiation of the same light source with that during IR light-driven $CO_2$ reduction. The FTO glass ($2.5 \times 1.5$ cm$^2$) deposited with materials was used as the photoelectrode, a Pt foil was used as the counter electrode, and Ag/AgCl electrode was used as the reference electrode. The three electrodes were inserted in a quartz cell filled with 0.2 M $Na_2SO_4$ electrolyte. The $Na_2SO_4$ electrolyte was purged with $CO_2$ for 1 h prior to the measurements. EIS was measured in the frequency of 1 to 1000 kHz.

## DFT calculation details

Density functional theory (DFT) calculations were carried out on a Vienna Ab initio Simulation Package (VASP)[39]. The exchange-correlation potential was described by the generalized gradient approximation (GGA) within the framework of Perdew-Burke-Ernzerhof (PBE) functional[40]. DFT-D3 method was employed to calculate the van der Waals (vdW) interaction[41]. The parameters of dipole correction were applied for the calculation of slab models. Electronic energies were computed with the tolerance of $1 \times 10^{-4}$ eV and total force of 0.01 eV/Å. A Monkhorst−Pack k-mesh of $2 \times 4 \times 1$ k-points was used in the structural relaxation, and a kinetic cutoff energy of 450 eV was adopted. The crystal lattice parameters of $Cu_4(SO_4)(OH)_6$ bulk are as follows: a = 13.087 Å, b = 9.865 Å, c = 6.015 Å ($\alpha = 90°$, $\beta = 103.33°$, $\gamma = 90°$) with the *P 1 21/a 1 (14)* space group. The $Cu_4(SO_4)(OH)_6$ slabs were modeled by the corresponding exposed surface along (100) direction with the thickness of single unit cell. The theoretical approach is based on the GGA with on-site Coulomb interaction parameter (GGA + U method), in which an effective U-J parameter of 5.2 eV was applied to improve the description Cu 3d states[42,43]. A vacuum space of 15 Å was inserted in z direction to avoid interactions between periodic images. All calculations were conducted under spin polarization.

The surface energy $E_s$ is the energy required to cleave a surface from the corresponding bulk crystal. It can be given by

$$E_s = 1/2A [E_s(unrelax) - N \times E_b] + 1/A [E_s(relax) - E_s(unrelax)] \quad (14)$$

where A is the area of the surface on the slab models, $E_s(unrelax)$ and $E_s(unrelax)$ represent the energy of the unrelaxed and relaxed surface slab models, respectively. N is the number of in the slab and $E_b$ is the energy of each atom in the bulk counterpart.

Adsorption energies $E_{adsorption}$ are given with reference to the isolated surface $E_{surface}$ relaxed upon removing the molecule from the unit cell using identical computational parameters and the energy of the molecule $E_{molecule}$.

$$E_{adsorption} = E_{molecule\ on\ surface} - E_{surface} - E_{molecule} \quad (15)$$

The computational hydrogen electrode (CHE)[44] model was used to calculate the Gibbs free energy change (ΔG) of $CO_2$ reduction reaction steps:

$$G = E_{DFT} + E_{ZPE} - TS \quad (16)$$

$$E_{ZPE} = \sum i\, 1/2hv_i \quad (17)$$

$$\Theta_i = hv_i/k \quad (18)$$

$$S = \sum i\, R\left[ln(1 - e^{-\Theta_i/T})^{-1} + \Theta_i/T(e^{\Theta_i/T} - 1)^{-1}\right] \quad (19)$$

where $E_{DFT}$ is the electronic energy calculated for specified geometrical structures, $E_{ZPE}$ is the zero-point energy, S is the entropy, h is

the Planck constant, v is the computed vibrational frequencies, Θ is the characteristic temperature of vibration, k is the Boltzmann constant, and R is the molar gas constant. For adsorbates, all 3 N degrees of freedom were treated as frustrated harmonic vibrations with negligible contributions from the catalysts' surfaces. Thus, free energy changes relative to an initial state can be represented by

$$\Delta G[COOH^*] = G[COOH^*] + 7 \times G[H^+ + e^-] - (G[*] + G[CO_2] + 8 \times G[H^+ + e^-]) \quad (20)$$

$$\Delta G[CO^*] = G[CO^*] + G[H_2O] + 6 \times G[H^+ + e^-] - (G[*] + G[CO_2] + 8 \times G[H^+ + e^-]) \quad (21)$$

$$\Delta G[* + CO] = G[*] + G[CO] + G[H_2O] + 6 \times G[H^+ + e^-] - (G[*] + G[CO_2] + 8 \times G[H^+ + e^-]) \quad (22)$$

$$\Delta G[CHO^*] = G[CHO^*] + G[H_2O] + 5 \times G[H^+ + e^-] - (G[*] + G[CO_2] + 8 \times G[H^+ + e^-]) \quad (23)$$

$$\Delta G[CH_2O^*] = G[CH_2O^*] + G[H_2O] + 4 \times G[H^+ + e^-] - (G[*] + G[CO_2] + 8 \times G[H^+ + e^-]) \quad (24)$$

$$\Delta G[CH_3O^*] = G[CH_3O^*] + G[H_2O] + 3 \times G[H^+ + e^-] - (G[*] + G[CO_2] + 8 \times G[H^+ + e^-]) \quad (25)$$

$$\Delta G[O^*] = G[O^*] + G[CH_4] + G[H_2O] + 2 \times G[H^+ + e^-] - (G[*] + G[CO_2] + 8 \times G[H^+ + e^-]) \quad (26)$$

$$\Delta G[OH^*] = G[OH^*] + G[CH_4] + G[H_2O] + G[H^+ + e^-] - (G[*] + G[CO_2] + 8 \times G[H^+ + e^-]) \quad (27)$$

$$\Delta G[* + CH_4 + 2H_2O] = G[*] + G[CH_4] + 2 \times G[H_2O] - (G[*] + G[CO_2] + 8 \times G[H^+ + e^-]) \quad (28)$$

$$G[H^+ + e^-] = 1/2G[H_2] - eU \quad (29)$$

where * is the substrate, U is the applied overpotential and *e* is the elementary charge. In this study, U = 0 V versus reversible hydrogen electrode.

## Data availability

The data that support the plots within this paper and other findings of this study are available from the corresponding author upon reasonable request. Source data are provided with this paper.

## Code availability

The code that supports the findings of this study is available from the corresponding author upon request.

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

## Acknowledgements

This work was financially supported by European Research Council (ERC) under the European Union's Horizon 2020 research and innovation program (grant agreement No 819698 and GrapheneCore3: 881603), Deutsche Forschungsgemeinschaft (COORNETs, SPP 1928 and CRC 1415: 417590517). The Supercomputing Center of Max Planck Computing & Data Facility (MPCDF) is acknowledged for computational support.

## Author contributions

X.F. and X.D.L. conceived the idea and co-wrote the paper. X.D.L., L.L., G.C., X.H.L., and X.C. carried out the sample synthesis, characterization and $CO_2$ reduction measurement. X.D.L. and L.L. discussed the catalytic process. C.N. helped with the BET characterizations. D.P. and M.L. guided the TEM and SEM characterizations. All the authors contributed to the overall scientific interpretation and edited the manuscript.

## Funding

## Competing interests

The authors declare no competing interests.
