## [Peer Review File · Nature Communications]

REVIEWER COMMENTS

Reviewer #1 (Remarks to the Author):

The manuscript by Feng et al reported the activated parity-forbidden electron transfer of d-d orbitals in transition metal complexes within ultrathin Cu-based hydroxalcalite-like hydroxy salts (CHHS) as an effective and general approach to promote IR light-driven CO₂ reduction.

The ultrathin Cu₄(SO₄)(OH)₆ nanosheets (CSON) are then synthesized as a typical catalyst to demonstrate this concept. The as-prepared CSON exhibit excellent activity for IR light-driven CO₂ reduction, with a production rate of 21.95 and 4.11 μmol g⁻¹ h⁻¹ for CO and CH₄

evolution, respectively. The concept has been demonstrated with good robust generality and is of great interest to the community. The overall manuscript has been organized with sound logic and enough details, while some minor problems, especially the real active sites and structure of the calcinated sample, need to be carefully checked and addressed before it can be accepted for publication.

1. When discussing the possibility for IR-driven CO₂ reduction, the arrows in Scheme 1 is not clear enough, please use another color to highlight the arrows.
2. When discussing the crystal structure and orientation of different catalysts, it is vital to get a direct view along different zone axis for different crystal structures, thus the crystal lattice axis within Figures S1 and S2 should be given.
3. By the way, when discussing the crystal structure and material thickness, the crystal lattice parameters and space group should be given for CSON.
4. It is mentioned that the defects are contributed by the removal of SO₄²⁻ after calcination, while looking at the TEM image in Figure 2a, significant large-sized holes can be observed, which might be due to the recrystallization or melting of the overall crystalline materials. The authors are suggested to carefully check the TD-DSC curves for the calcinated sample and the composition and crystallographic points of view to double-check the composition within the calcined sample. From PXRD in Figure S7, it is significant that the first diffraction peak is much weaker, while a significant peak shift has been observed within high 2θ positions. Please double-check the structure of materials under this circumstance.
5. When discussing ultrathin and metallic semiconducting photocatalysts for CO₂ reduction, except copper-based material, a lot of well-reported Bismuth-based materials for CO₂ reduction, such as Bismuth oxides, Bismuth carbonates, Bi₁₉Br₃S₂₇, etc. reported in the literature should not be excluded within the discussion.
6. For the Tauc plots in figure 3b and 3c, the first two edge position values are quite similar to each other, while the VBM has been concluded to be significantly different, how much do you confident about these differences? Any approaches to double-confirm these conclusions?
7. During the material synthesis, why has CTAB been used in the synthesis? Will that influence the catalytic properties? Does that contribute to catalytic product generation?

8. The stability of the catalyst is doubted, as shown in Figure S21, significant impurity diffraction peaks have been observed within the samples after photocatalysis, under this circumstance, it is not clear yet what is the active catalytic center, the author is suggested double confirm the stability and real active sites within the catalysts.

Reviewer #2 (Remarks to the Author):

In this manuscript, Li et al. prepared a series of 2D ultrathin Cu-based hydrotalcite-like hydroxy salts. For the first time, they applied these 2D materials for IR light-driven CO₂ photoreduction and achieved excellent performance. More impressively, they proposed a brand-new “Activated parity-forbidden transition of d-d orbitals” mechanism for electron transfer under low-energy IR light. This approach will inspire the development of highly efficient photocatalysts, especially for IR light-driven catalytic system. Various experimental characterizations were performed to reveal carrier dynamic behaviors and the evolution of catalysts. Besides, the theoretical DFT calculations were carried out to further explain the catalytic processes and reaction mechanism. The experimental results and the demonstration are highly self-consistent with each other, and the novelty is very sufficient. So, I would like to recommend its publication in “Nature Communications” after minor revisions. Some tiny issues should be concerned as follows:

Comments and suggestions:

1. Is it possible to define the band structure of these 2D materials in this paper by some experimental methods?
2. What is advantage of the 2D structure of Cu-based hydrotalcite-like hydroxy salts for IR light-driven CO₂ photoreduction?
3. In figure 3d, the value of VBM from the UPS seems different with that showed in figure 3f. Is that wrong here?
4. Since the IR light normally causes a lot of heat, how to exclude the affection of that?
5. Some icon is missing, like Supplementary Figure 23c, and the whole Supplementary Figure 37. Please check it.
6. English expressions in some places need to be carefully polished.

Reviewer #3 (Remarks to the Author):

This paper demonstrated a new strategy on regulation of 2D semiconductors photocatalysts, which enables them with promoted IR light-driven photocatalytic CO₂ reduction. Their findings, activated parity-forbidden transition of d-d orbitals, are relevant to pave the way towards the rational design and band engineering of photocatalysts. Due to the general utility of this method for photocatalysis, I expect this work to be of interest to the broad readership of Nature Communications. I recommend the acceptance of this work after minor revision.

(1) Why the authors choose the ultrathin Cu based photocatalysts as models, and what is the principle?

(2) The disadvantages of IR-light-driven photocatalysis should be discussed, as we know that the visible-light-driven photocatalysis has been widely studied.

(3) According to the synchrotron-radiation photoemission spectroscopy in Fig. 3d and Supplementary Figure 12a, the obtained valence bands of c-CSON and p-CSON do not agree with the position displayed in the corresponding energy band structure scheme, is that correct?

(4) The pre-edge of XANES in Figure 2e should be enlarged.

(5) Since there is oxygen generated during the reaction, would the samples be oxidized somehow?

Reviewer #4 (Remarks to the Author):

I have thoroughly read the manuscript of “Activated parity-forbidden transition of d-d orbitals for infrared light-driven CO₂ reduction” by Li et al. Infrared light-driven CO₂ reduction is difficult for the current photocatalysts. In this work, the authors proposed a novel approach of “d-d electron transfer” to realize the efficient IR light CO₂ reduction. Then, they used DFT simulations and predicted a series Cu-based photocatalysts that can realize this purpose. Interestingly, the final photocatalytic performance was improved. The key points of this work can be concluded as (1) the novel approach to regulate the band and electron feature of photocatalyst; (2) the universality of this approach for various photocatalysts; (3) the obtained performance feedbacks were greatly improved. Thus, I would recommend to publish this work “Nature Communications” after minor revisions. The following questions and suggestions should be further considered:

Comment 1: Two kinds of Cu sites “tetrahedral” and “octahedral” coexist in the photocatalysts, and the band structures were changed after calcining. As the results, the reactivity was improved with the increasement of “tetrahedral” sites. How about the selectivity of CO and CH₄? The results of selectivity should be calculated.

Comment 2: In this work, the 2D hydrotalcite-like hydroxy copper salts were well regulated. Can the authors predict what kind of materials might be also optimized with this method, and what characteristics are required with these materials.

Comment 3: Various photocatalysts of CMON, CPON and CCON were investigated, while the performance feedbacks were different, such as the activity, selectivity and the ratio of CO vs CH₄. The authors should explain the origins of these differences.

Comment 4: Line 327 in the main text, there is no definition of the symbols CNON, CPON and CCON samples, which should be added.

Comment 5: In scheme 1a, the position of CB symbol is not clear. In Supplementary figure S23, the figure (c) is missed.

Comment 6: In figure 3d, the VB from the UPS is 3.09 eV, it seems not the same position with that showed in figure 3f. Why?

Responses to the Reviewers' comments and a summary of the changes made to the manuscript: NCOMMS-23-08584-T. We would like to thank all the reviewers for the insightful comments and suggestions, and for their time in helping us to improve this manuscript.

And we acknowledge the Reviewers' positive comments that “The overall manuscript has been organized with sound logic and enough details”, “I would like to recommend its publication in “Nature Communications” after minor revisions” and “I recommend the acceptance of this work after minor revision”.

Point-to-point Response to Reviewer #1

Overall comments: The manuscript by Feng et al reported the activated parity-forbidden electron transfer of d-d orbitals in transition metal complexes within ultrathin Cu-based hydrotalcite-like hydroxy salts (CHHS) as an effective and general approach to promote IR light-driven CO₂ reduction. The ultrathin Cu₄(SO₄)(OH)₆ nanosheets (CSON) are then synthesized as a typical catalyst to demonstrate this concept. The as-prepared CSON exhibit excellent activity for IR light-driven CO₂ reduction, with a production rate of 21.95 and 4.11 μmol g⁻¹ h⁻¹ for CO and CH₄ evolution, respectively. The concept has been demonstrated with good robust generality and is of great interest to the community. The overall manuscript has been organized with sound logic and enough details, while some **minor problems**, especially the real active sites and structure of the calcinated sample, need to be carefully checked and addressed before it can be accepted for publication.

Overall response: We greatly appreciate the Reviewer's positive comments and constructive suggestions guiding the revision of our manuscript.

Comment 1: When discussing the possibility for IR-driven CO₂ reduction, the arrows in Scheme 1 is not clear enough, please use another color to highlight the arrows.

Response: We appreciate the Reviewer's suggestions. We **have changed** the color of the arrows in Scheme 1 to red to highlight them. The corresponding images are displayed in Fig. N1 as follows:

Fig. N1 (Scheme 1) | Possible electron transition mechanisms for IR light-driven CO₂ reduction. (a) Metallic catalysts (the energy bands below the Fermi level with fully occupied electrons are regarded as CB and the first band above the Fermi level without electron occupation is called to B₁); (b) semiconductors with the intermediate band; (c) hydrotalcite-like hydroxy salts with d-d orbital transition. Green ball: electrons; White ball: holes; Red lightning: IR light irradiation; Red arrows: electron transfer pathway; CB: conduction band; VB: valence band; B₁: Lowest

empty band; IB: Intermediate band; — Fermi level.

Comment 2: When discussing the crystal structure and orientation of different catalysts, it is vital to get a direct view along different zone axis for different crystal structures, thus the crystal lattice axis within Figures S1 and S2 should be given.

Response: Following the suggestion, we **have added** the corresponding crystal lattice axis into Supplementary Figure 1 (Fig. N2) and Supplementary Figure 2 (Fig. N3) as follows:

Fig. N2 (Supplementary Figure 1) | The slab models with the thickness of a single unit cell and the corresponding theoretical surface energy for CSON. (a) [100] facet, (b) [010] facet and (c) [001] facet.

Fig. N3 (Supplementary Figure 2) | Theoretical simulations of 2D CHHS. Theoretical slab models, the density of states and energy band structures for (a)-(c) CCON, (d)-(f) CPON and (g)-(i) CNON. Black and red lines in (c), (f) and (i) represent the spin up and spin down energy band respectively.

Comment 3: *By the way, when discussing the crystal structure and material thickness, the crystal lattice parameters and space group should be given for CSON.*

Response: We **have added** the corresponding description about the crystal lattice parameters and space group of CSON at the “DFT calculation details” part in the revised manuscript as follows:

“The crystal lattice parameters of $\text{Cu}_4(\text{SO}_4)(\text{OH})_6$ bulk are as follows: $a = 13.087 \text{ \AA}$, $b = 9.865 \text{ \AA}$, $c = 6.015 \text{ \AA}$ ($\alpha = 90^\circ$, $\beta = 103.33^\circ$, $\gamma = 90^\circ$) with the $P 1 21/a 1 (14)$ space group.”

Comment 4: *It is mentioned that the defects are contributed by the removal of SO_4^{2-} after calcination, while looking at the TEM image in Figure 2a, significant large-sized holes can be observed, which might be due to the recrystallization or melting of the overall crystalline materials. The authors are suggested to carefully check the TG-DSC curves for the calcinated sample and the composition and crystallographic points of view to double-check the composition within the calcined sample. From PXRD in Figure S7, it is significant that the first diffraction peak is much weaker, while a significant peak shift has been observed within high 2theta positions. Please double-check the structure of materials under this circumstance.*

Response: We appreciate the Reviewer's comments and suggestions. For the holes appearing on the surface of c-CSON, it can be attributed to the formation of a large number of surface SO_4 defects, but from the TEM images, the overall flake morphology of the sample did not change. There have been numerous reports confirming that surface defects can lead to the generation of holes on the surface of 2D structures (*Surfaces and Interfaces* 2021, 23, 100979; *Mater. Chem. Front.* 2017, 1, 2065-2077; *J. Alloys Compd.* 2018, 737, 113-121.).

According to the Reviewer's suggestion, we also carried out the experiments of thermogravimetry-differential scanning calorimetry (TG-DSC) and thermogravimetry-differential thermal gravity (TG-DTG) curves for the pristine and calcinated samples as shown in Fig. N4. These two samples have very similar TG-DSC and TG-DTG curves, confirming that the crystal phase and composition of the two samples are almost identical before and after calcination. From the TG and DTG curves, it can be seen that both samples only have a small amount of mass loss at 300°C , which may be due to surface defects or the loss of crystalline water in the samples, without the occurrence of sample decomposition and reconstruction. The samples decompose completely into CuO at around 770°C , confirming the stability of the samples. Meanwhile, the DSC curves of both p-CSON and c-CSON show that there is no recrystallization or dissolution process before 300°C , and only a recrystallization peak occurs at $552\text{-}560^\circ\text{C}$, followed by a melting peak at $717\text{-}718^\circ\text{C}$. Therefore, according to the results of TG-DSC and TG-DTG curves, we can confirm that there is no change in the crystal phase and composition of our samples during the calcination process at 300°C , which is consistent with the results of XRD patterns and FTIR spectra.

We **have added** the corresponding description and results of TG-DSC and TG-DTG curves into the revised Supplementary Information.

Fig. N4 (Supplementary Figure 6) | (a) TG-DSC and (b) TG-DTG for p-CSON. (c) TG-DSC and (d) TG-DTG for c-CSON.

For PXRD measurement, the overall intensity of the diffraction peak is highly dependent on the quality of the samples and testing conditions. To confirm that, we conducted multiple PXRD measurements with different amounts of p-CSON again, as shown in Fig. N5 below. Additionally, we only used the comparison of the diffraction peak position between the sample and standard reference cards to determine the phase of the samples. And we did not observe “*a significant peak shift*” as mentioned by the Reviewer. Perhaps there is a slight difference in the overall peak position before and after calcination, which could be attributed to the instrument's testing deviation. To avoid any misunderstandings, we **have also replaced** the spectral image with a clearer one (Fig. N6) in the revised Supplementary Information.

Fig. N5 | XRD patterns of p-CSON with different amounts of sample.

Fig. N6 (Supplementary Figure 8a) | XRD patterns of p-CSON and c-CSON.

Comment 5: When discussing ultrathin and metallic semiconducting photocatalysts for CO_2 reduction, except copper-based material, a lot of well-reported Bismuth-based materials for CO_2 reduction, such as Bismuth oxides, Bismuth carbonates, $\text{Bi}_{19}\text{Br}_3\text{S}_{27}$, etc. reported in the literature should not be excluded within the discussion.

Response: We appreciate the Reviewer's comments and suggestions. We **have added** the discussion and references about Bismuth-based materials for CO_2 photoreduction in the manuscript as follows:

“Besides, some bismuth-based catalysts, like metallic $\text{Bi}_{19}\text{Br}_3\text{S}_{27}$ nanowires [J. Am. Chem. Soc. 2021, 143, 6551-6559], have also been reported to exhibit excellent activity for CO_2 reduction under near-infrared (NIR) light irradiation.”

Comment 6: For the Tauc plots in figure 3b and 3c, the first two edge position values are quite similar to each other, while the VBM has been concluded to be significantly different, how much do you confident about these differences? Any approaches to double-confirm these conclusions?

Response: Thanks for the Reviewer’s concerns. First of all, we are very confident in the results of the valence band maxima (VBM) we got for p-CSON and c-CSON. The VBM positions were determined and confirmed by multiple UPS tests, which are mainly related to the work function, Fermi level and secondary electron cutoff of the samples. The Tauc plots are only used to identify the position of conduction band maxima (CBM) or intermediate bands (*Nat. Energy* 2021, 6, 388-397.). As for the Reviewer’s concerns of “the first two edge position values are quite similar to each other”, the reason could be attributed to the difference of the middle empty d bands in the p-CSON and c-CSON as shown in Fig. N7. The first two absorption edges in Fig. N7a and c are generated by the electron transfer processes of I and II in Fig. N7b and d. The suitable position of the middle empty d bands could result in a similar low-energy absorption edge.

Fig. N7 | Tauc plots and band structure for p-CSON (a, b) and c-CSON (c, d).

As for the Reviewer’s second question, to define the band structure of catalysts, Mott-Schottky plot is also a widely-used approach. To make our results more credible and rigorous, we further carried out the corresponding experiments as displayed in Fig. N8. The flat potentials (E_{fb}) of p-CSON and c-CSON are -0.56 V and -0.40 V vs Ag/AgCl (-0.36 V and -0.2 V vs NHE), respectively. Since the conduction band position (E_{CB}) is more negative by ca. -0.1 V than E_{fb} for n-type semiconductors (*Nat. Commun.* 2022, 13, 2964; *Nat. Commun.* 2022, 13, 4900; *Angew. Chem.Int. Ed.* 2023, 62, e202216613), the E_{CB} of p-CSON and c-CSON can be calculated to be -0.46 V and -0.30 V vs NHE at pH = 0, respectively. According to the intrinsic band gap obtained by the UV-vis-NIR diffuse reflectance spectra, we could also get the valence band position (E_{VB}) of p-CSON and c-CSON to 2.71 V and 2.43 V, respectively. The obtained band structures by the Mott-Schottky plots are quite similar to that of UPS in our work, strongly confirming the credibility of our results. We **have added** the test methods and results of Mott-

Schottky plots into the revised Supplementary Information.

Fig. N8 (Supplementary Figure 15) | Mott-Schottky plots of p-CSON and c-CSON. The Mott-Schottky plots were performed on the CHI 660E electrochemical workstation via a standard three-electrode system in 0.2 M Na₂SO₄ solution which contains a working electrode, a platinum plate as counter electrode. The catalyst (5 mg) was dispersed into a solution of 25 μ L 5 wt% Nafion and 0.5 mL isopropanol. Then the resulting mixture (0.2 mL) was deposited onto the surface of FTO and left in the air for drying to prepare the working electrode. The Mott-Schottky plots were recorded at frequency of 1000 Hz.

Comment 7: *During the material synthesis, why has CTAB been used in the synthesis? Will that influence the catalytic properties? Does that contribute to catalytic product generation?*

Response: CTAB was used as a surfactant to ensure the formation of 2D ultrathin configuration of p-CSON in our work, in which CTAB could be adsorbed in a specific plane ([100] plane in our work), further promoting the growth of crystals in two-dimensional direction. This surfactant-assisted approach has been widely used in the synthesis of 2D materials to form ultrathin flake-like structure (*Nature* 2016, 529, 68-71; *J. Am. Chem. Soc.* 2014, 136, 6826-6829). And we also synthesized the Cu₄(SO₄)(OH)₆ without CTAB. As displayed in Fig. N9, the TEM image shows that the obtained Cu₄(SO₄)(OH)₆ is at least 15 nm, much thicker than samples synthesized with CTAB.

Fig. N9 | TEM image of $\text{Cu}_4(\text{SO}_4)(\text{OH})_6$ synthesized without CTAB.

As for the influence of CTAB on catalytic properties and product generation, we excluded their influence on catalytic processes in our work. We got the final catalysts of p-CSON and c-CSON after washing the samples with ethanol and deionized (DI) water at least six times. We could not identify any residual of CTAB on the surface of our samples. For instance, we performed FTIR spectra in our manuscript to confirm that. The characteristic peaks of CTAB are reported at 2918 and 2850 cm^{-1} caused by the C-H stretching vibration (*Langmuir* 2015, 31, 817-823; *RSC Adv.* 2015, 5, 100142-100146; *J. Petrol Explor. Prod. Technol.* 2018, 8, 597-606), but the FTIR spectra of p-CSON and c-CSON in Supplementary Figure 8b show that there is no any peak of CTAB detected. Therefore, we have ruled out the presence of CTAB before conducting catalytic experiments using p-CSON and c-CSON. To make it clearer, we **have added** the FTIR spectra of CTAB (Fig. N10) and corresponding explanations in the revised Supplementary Information as follows:

Fig.N10 (Supplementary Figure 8b) | FTIR spectra for p-CSON and c-CSON. One can clearly see that the phase and structures are well retained after the calcination. And both p-CSON and c-CSON possess clean surface without CTAB.

Comment 8: *The stability of the catalyst is doubted, as shown in Figure S21, significant impurity diffraction peaks have been observed within the samples after photocatalysis, under this circumstance, it is not clear yet what is the active catalytic center, the author is suggested*

double confirm the stability and real active sites within the catalysts.

Response: We apologize for any confusion caused to the Reviewer. We have found a sudden diffraction peak appearing around 12 degrees in the XRD pattern of p-CSON in Supplementary Figure 21c (in the original manuscript), which we believe is caused by instrument disturbance. To further investigate, we have conducted multiple XRD measurements of p-CSON with different amounts after catalysis. As shown in the Fig. N11 below, we found that the disturbance at this position disappeared. To avoid any misunderstandings, we **have replaced** the image with a clearer one (Fig. N12) in the revised Supplementary Information.

Fig. N11 | PXR patterns of p-CSON after catalysis with different amounts of sample.

Fig. N12 (Supplementary Figure 24e) | PXR patterns of p-CSON and c-CSON after IR light catalysis.

To double confirm the stability, we further carried out the FTIR spectra and HRTEM of p-CSON and c-CSON after the continuous 96 h test for photocatalytic CO₂ under IR light irradiation. As displayed in Fig. N13c and d, the crystal diffraction stripes, interplanar spacing, and exposed crystal faces of p-CSON and c-CSON after catalysis are basically unchanged compared to that before catalysis. Meanwhile, the FTIR spectra (Fig. N13f) show that the characteristic infrared peaks of p-CSON and c-CSON after catalysis remain consistent with those before catalysis, without any significant changes. All of the above strongly confirm that both p-CSON and c-CSON have good stability. We **have added** the below FTIR spectra and

Fig. N13 (Supplementary Figure 24) | Characterizations for p-CSON and c-CSON after the continuous 96 h test for photocatalytic CO₂ under IR light irradiation. TEM images for (a) p-CSON and (b) c-CSON. HRTEM images for (c) p-CSON and (d) c-CSON. XRD patterns (e) and FTIR spectra (f) for p-CSON and c-CSON.

We agree that the active catalytic center is an important research topic for any catalytic reaction. However, so far, it is still difficult for us to directly observe the real catalytic site in experiments. Transition metals, due to their half-filled d orbitals and high charge density, can not only provide additional orbitals that bond with reactants but also participate in catalytic reactions by providing electrons. Therefore, they are widely regarded as catalytic reaction centers (*Nat. Rev. Mater.* 2022, 7, 503-521; *Nat. Commun.* 2019, 10, 3169; *Nature* 2020, 586, 549-554.). In particular, Cu has been extensively reported to be an active center for CO₂ reduction reactions (*J. Am. Chem. Soc.* 2021, 143, 2984-2993; *Nat. Energy* 2019, 4, 957-968.). In our work, the

charge density distribution in Supplementary Figure 30 confirms that the rich charge density of Cu metal can promote the formation and adsorption of intermediates, consistent with the calculated adsorption energy of COOH* intermediates in Supplementary Figure 31. The charge density difference (CDD) in Supplementary Figure 32 also shows strong electron transfer and bonding between Cu sites and the reaction intermediate of COOH* species. Meanwhile, comparing p-CSON with the defective c-CSON, we found that the low-coordinated Cu (tetrahedral Cu) significantly increased in c-CSON. Consequently, the product yield of c-CSON also increased, but the product selectivity did not change significantly, suggesting that c-CSON and p-CSON have similar active centers, only with different contents of catalytic sites. Therefore, we speculate that the tetrahedral Cu is a possible catalytic active site. We **have added** the above discussion in the revised manuscript as follows:

“Transition metals are noteworthy for their half-filled d orbitals and high charge density, which allow them to not only provide additional orbitals for bonding with reactants but also participate in catalytic reactions by providing electrons [Nat. Commun. 2019, 10, 3169]. In particular, Cu site has been extensively studied as an active center for CO₂ reduction reactions [Nat. Energy 2019, 4, 957-968.]. In comparing p-CSON with defective c-CSON, we observed a significant increase in low-coordinated Cu (tetrahedral Cu) in c-CSON. As a result, the product yield also increased, but the product selectivity remained largely unchanged. This suggests that c-CSON and p-CSON have similar active centers, with differences only in the amount of catalytic sites. Therefore, tetrahedral Cu is regarded as the active site for CO₂ reduction.”

Point-to-point Response to Reviewer #2

Overall comments: *In this manuscript, Li et al. prepared a series of 2D ultrathin Cu-based hydrotalcite-like hydroxy salts. For the first time, they applied these 2D materials for IR light-driven CO₂ photoreduction and achieved excellent performance. More impressively, they proposed a brand-new “Activated parity-forbidden transition of d-d orbitals” mechanism for electron transfer under low-energy IR light. This approach will inspire the development of highly efficient photocatalysts, especially for IR light-driven catalytic system. Various experimental characterizations were performed to reveal carrier dynamic behaviors and the evolution of catalysts. Besides, the theoretical DFT calculations were carried out to further explain the catalytic processes and reaction mechanism. The experimental results and the demonstration are highly self-consistent with each other, and the novelty is very sufficient. So, I would like to recommend its publication in “Nature Communications” after minor revisions.*

Overall response: Thank a lot for the Reviewer’s positive comments and kind suggestions.

Comment 1: *Is it possible to define the band structure of these 2D materials in this paper by some experimental methods?*

Response: To our knowledge, angle-resolved photoemission spectroscopy (ARPES) can be used to measure the band structure of relatively large single-crystal materials. However, so far there is no experimental technique available for characterizing the band structure of powder

samples or nanomaterials with small sizes. Therefore, we can only infer the band structure of c-CSON and p-CSON indirectly by combining UPS and UV-vis-NIR diffuse reflectance spectra.

Comment 2: *What is advantage of the 2D structure of Cu-based hydrotalcite-like hydroxy salts for IR light-driven CO₂ photoreduction?*

Response: In bulk materials, the IR light-excited active electrons easily lose their activity due to their inability to rapidly transfer to the catalytic sites and participate in reactions by recombining with holes. However, in 2D structure, thanks to their ultra-thin configuration in one direction, the electron-hole pairs can be rapidly separated and transferred to the catalytic surface to participate in reactions, thereby suppressing carrier recombination and improving catalytic activity. On the other hand, the rich exposed surface of 2D structures provides more low-coordinated active sites, which not only improves the light absorption efficiency but also enhances the catalytic performance. We have already had a relevant discussion in the “Introduction” section.

Comment 3: *In figure 3d, the value of VBM from the UPS seems different with that showed in figure 3f. Is that wrong here?*

Response: Thanks for the Reviewer's concerns. The SRPES valence-band spectra were measured using synchrotron-radiation light as the excitation source with a photon energy of 40.00 eV (performed at the Catalysis and Surface Science Endstation at the BL11U beamline of the National Synchrotron Radiation Laboratory), while the value of the valence band maxima (VBM) shown in Fig. 3d and Supplementary Figure 13a are referenced to the Fermi level determined from Au, called as E_{edge} shown in Fig. N14. The valence band maxima (VBM) of the samples referenced to Normal Hydrogen Electrode (NHE) can be obtained according to the following equations (*Angew. Chem. Int. Ed.* 2019, 58, 11791-11795; *Nat. Energy* 2021, 6, 388-397):

$$\begin{aligned}\Phi &= h\nu - E_{\text{cutoff}} \\ E_{\text{VBM}} &= E_{\text{edge}} + \Phi - 4.5 \quad (\text{vs NHE, pH} = 0)\end{aligned}$$

where Φ is the work function, $h\nu$ is the photon energy of the excitation source, E_{cutoff} is the energy of secondary electron cutoff, and E_{VBM} is the valence band maxima of samples vs NHE at pH = 0. According to the above, we can get the E_{VBM} of c-CSON (2.42 V in Fig. 3f) and p-CSON (2.70 V in Supplementary Figure 13c).

Fig. N14 (Supplementary Figure 14) | Schematic illustration of the energy level information from UPS.

To make it clearer, we **have added** the above discussion to the revised manuscript and modified the corresponding expression.

Comment 4: *Since the IR light normally causes a lot of heat, how to exclude the effect of that?*

Response: Thanks for the Reviewer's concerns. To exclude the effect of IR light-induced heat, a water-cooled gas-solid reaction system (Supplementary Fig. 17a-b) is designed and applied in this work to avoid the effects of thermocatalysis, in which the quartz boat floats directly on the water. The local temperature is monitored in real-time by an infrared thermal imaging camera. The average temperature of CSON-based thin film during IR light irradiation has proved to be almost unchanged during the catalysis as shown in Supplementary Fig. 17c-h.

Comment 5: *Some icon is missing, like Supplementary Figure 23c, and the whole Supplementary Figure 37. Please check it.*

Response: For Supplementary Figure 23 and 37 (in the original manuscript), we **have added** the corresponding icon in the revised Supporting Information. We are sorry for our carelessness.

Comment 6: *English expressions in some places need to be carefully polished.*

Response: According to Reviewer's kind suggestion, we **have carefully edited** the mistyping or grammatical errors in the revised manuscript. And we also invited Prof. Lei Fang from Texas A&M University for constructive suggestions to English modification of this work. He has been studied and worked in the United States for many years. In this work, He helps to do a modification on our original manuscript in order to eliminate possible grammatical errors and make it more fluent and understandable.

Point-to-point Response to Reviewer #3

Overall comments: *This paper demonstrated a new strategy on regulation of 2D semiconductors photocatalysts, which enables them with promoted IR light-driven photocatalytic CO₂ reduction. Their findings, activated parity-forbidden transition of d-d*

orbitals, are relevant to pave the way towards the rational design and band engineering of photocatalysts. Due to the general utility of this method for photocatalysis, I expect this work to be of interest to the broad readership of Nature Communications. **I recommend the acceptance of this work after minor revision.**

Overall response: We appreciate the Reviewer's positive comments and kind suggestions.

Comment 1: *Why the authors choose the ultrathin Cu based photocatalysts as models, and what is the principle?*

Response: In recent years, Cu-based catalysts have been shown to exhibit high activity for CO₂ reduction, with diverse product selectivity and high catalytic performance. In addition, for photocatalytic CO₂ reduction, rapid separation of charge carriers can improve their utilization efficiency, further enhancing catalytic activity. The 2D configuration reduces the distance for charge carriers to migrate to the surface to the atomic scale in one dimension, thus promoting charge carrier separation and effectively participating in catalytic reactions, leading to improved catalytic activity. Therefore, ultrathin Cu-based photocatalysts are a promising candidate for photocatalytic CO₂ reduction.

The basic principle for selecting catalysts for photocatalytic CO₂ reduction are as follows :

- (1) Excellent light absorption ability;
- (2) Fast carrier separation and transport capability;
- (3) Reasonable conduction and valence band position for CO₂ reduction and H₂O oxidation reaction.

Certainly, the abovementioned are only the prerequisites for realizing effective CO₂ photoreduction reaction. The active sites, adsorption energy for reactants and intermediate and other characteristics also need to be considered for the specific catalytic reactions.

Comment 2: *The disadvantages of IR-light-driven photocatalysis should be discussed, as we know that the visible-light-driven photocatalysis has been widely studied.*

Response: Thanks for the Reviewer's suggestion. The energy of IR light is low, typically below 1.55 eV, making it difficult to excite the catalyst's electrons to participate in the catalytic reaction. Additionally, IR light can generate heat easily. If the catalytic system does not include condensate water, it may cause local high temperatures, which can deactivate the catalyst. However, for the solar spectrum, IR light has a high percentage, accounting for about 50%. Improving the utilization of infrared light is beneficial for increasing the overall utilization of solar energy. Studying IR light-driven catalysts can explore efficient infrared photocatalytic reactions, and coupling high-performance infrared catalysts with ultraviolet or visible catalysts can promote the development of all-spectrum photocatalysts, which is of great significance. We **have added** the corresponding discussion in the revised manuscript as follows:

“Recently, the IR light-driven redox reactions have gained widespread attention among researchers. Despite the low energy of IR light and its tendency to generate localized heat, its relatively high proportion in the solar spectrum (ca. 50%) has prompted people to explore ways to utilize it.”

Comment 3: *According to the synchrotron-radiation photoemission spectroscopy in Fig. 3d*

and Supplementary Figure 12a, the obtained valence bands of c-CSON and p-CSON do not agree with the position displayed in the corresponding energy band structure scheme, is that correct?

Response: Thanks for the Reviewer's concerns. The SRPES valence-band spectra were measured using synchrotron-radiation light as the excitation source with a photon energy of 40.00 eV (performed at the Catalysis and Surface Science Endstation at the BL11U beamline of the National Synchrotron Radiation Laboratory), while the value of the valence band maxima (VBM) shown in Fig. 3d and Supplementary Figure 13a are referenced to the Fermi level determined from Au, called as E_{edge} shown in Fig. N14. The valence band maxima (VBM) of the samples referenced to Normal Hydrogen Electrode (NHE) can be obtained according to the following equations (*Angew. Chem. Int. Ed.* 2019, 58, 11791-11795; *Nat. Energy* 2021, 6, 388-397):

$$\Phi = h\nu - E_{\text{cutoff}}$$

$$E_{\text{VBM}} = E_{\text{edge}} + \Phi - 4.5 \quad (\text{vs NHE, pH} = 0)$$

where Φ is the work function, $h\nu$ is the photon energy of the excitation source, E_{cutoff} is the energy of secondary electron cutoff, and E_{VBM} is the valence band maxima of samples vs NHE at pH = 0. According to the above, we can get the E_{VBM} of c-CSON (2.42 V in Fig. 3f) and p-CSON (2.70 V in Supplementary Figure 13c).

Fig. N14 (Supplementary Figure 14) | Schematic illustration of the energy level information from UPS.

To make it clearer, we **have added** the above discussion to the revised manuscript and modified the corresponding expression.

Comment 4: *The pre-edge of XANES in Figure 2e should be enlarged.*

Response: We **have revised** the Figure 2e and put the enlarged pre-edge of XANES (Fig. N15) into the insert images as follows:

Fig. N15 (Fig. 2e) | Cu K-edge XANES spectra, the insert is a zoomed-in view of the part of the green circle.

Comment 5: *Since there is oxygen generated during the reaction, would the samples be oxidized somehow?*

Response: Thanks for the Reviewer's concerns. We extensively characterized the post-catalysis samples of p-CSON and c-CSON. As displayed in Supplementary Figure 24 in the revised Supplementary Information, TEM and HRTEM images confirm that the morphology, crystal diffraction stripes, interplanar spacing, and exposed crystal faces of p-CSON and c-CSON after catalysis are basically unchanged compared to that of samples before catalysis. XRD patterns demonstrate that the phase and crystal structure of p-CSON and c-CSON also remain unchanged before and after catalysis. Meanwhile, the FTIR spectra show that the characteristic infrared peaks of p-CSON and c-CSON after catalysis remain consistent with those before catalysis, without any significant changes. All of the above strongly confirms that both p-CSON and c-CSON have good stability, and no obvious oxidation components are detected.

Point-to-point Response to Reviewer #4

Overall comments: *I have thoroughly read the manuscript of "Activated parity-forbidden transition of d-d orbitals for infrared light-driven CO₂ reduction" by Li et al. Infrared light-driven CO₂ reduction is difficult for the current photocatalysts. In this work, the authors proposed a novel approach of "d-d electron transfer" to realize the efficient IR light CO₂ reduction. Then, they used DFT simulations and predicted a series Cu-based photocatalysts that can realize this purpose. Interestingly, the final photocatalytic performance was improved. The key points of this work can be concluded as (1) the novel approach to regulate the band and electron feature of photocatalyst; (2) the universality of this approach for various photocatalysts; (3) the obtained performance feedbacks were greatly improved. Thus, **I would recommend to publish this work "Nature Communications" after minor revisions.***

Overall response: We appreciate the Reviewer for the positive comments and kind suggestions.

Comment 1: *Two kinds of Cu sites "tetrahedral" and "octahedral" coexist in the*

photocatalysts, and the band structures were changed after calcinating. As the results, the reactivity was improved with the increasement of “tetrahedral” sites. How about the selectivity of CO and CH₄? The results of selectivity should be calculated.

Response: Thanks for the Reviewer’s concerns. We **have calculated** the product and electron selectivity of CO and CH₄ for both p-CSON and c-CSON. The selectivity for p-CSON and c-CSON did not change significantly, indicating that both the p-CSON and c-CSON have similar active sites and reaction mechanisms. And the corresponding discussion and methods **have been provided** in the revised manuscript as below:

“The product selectivity of CO and CH₄ is calculated to 84.2 % and 15.8 % for c-CSON, 85.7 % and 14.3 % for p-CSON; while the electron selectivity is 57.2 % and 42.8 % for c-CSON, 59.9 % and 40.1 % for p-CSON.”

“The product selectivity for CO₂ reduction to CO and CH₄ has been calculated using the following equation:

$$\text{Product selectivity of CO (\%)} = [n(\text{CO})] / [n(\text{CH}_4) + n(\text{CO})] \times 100\%$$

$$\text{Product selectivity of CH}_4 \text{ (\%)} = [n(\text{CH}_4)] / [n(\text{CH}_4) + n(\text{CO})] \times 100\%$$

The electron selectivity for CO₂ reduction to CO and CH₄ (2 e⁻ for CO, 8 e⁻ for the formation of CH₄) has been calculated using the following equation:

$$\text{Electron selectivity of CO (\%)} = [2n(\text{CO})] / [8n(\text{CH}_4) + 2n(\text{CO})] \times 100\%$$

$$\text{Electron selectivity of CH}_4 \text{ (\%)} = [8n(\text{CH}_4)] / [8n(\text{CH}_4) + 2n(\text{CO})] \times 100\%$$

where n(CH₄) and n(CO) are the amounts of produced CH₄ and CO.”

Comment 2: In this work, the 2D hydrotalcite-like hydroxy copper salts were well regulated. Can the authors predict what kind of materials might be also optimized with this method, and what characteristics are required with these materials.

Response: It is really a good question. We consider that a lot of hydrotalcite-like hydroxy metal salts might be able to optimize with this method since they have a very similar structure. Actually, we found that some similar materials, like basic nickel carbonate (*Eur. J. Inorg. Chem.* 2015, 5913-5920) and Cu₂(OH)₃Cl (*Chem.Eur.J.*2015,21,13583-13587), exhibit IR light absorption ability and catalytic activities. And other transition metal complexes with tetrahedral or octahedral coordination may also have huge potential to be optimized by this approach. For IR light-driven CO₂ reduction, several characteristics should be required as follows:

- (1) Excellent IR light absorption ability;
- (2) Fast carrier separation and transport capability;
- (3) Efficient electron transfer pathway under IR light irradiation, the excited electrons should be transferred to the position for CO₂ reduction and the remaining hole should be the position for H₂O oxidation reaction.

Certainly, the abovementioned are only the prerequisites for realizing IR light-driven CO₂ reduction reactions. The active sites, adsorption energy for reactants and intermediate and other characteristics also need to be considered for the specific catalytic reactions. A lot of work are still needed to carry out to further explore the intrinsic mechanism for IR light-driven CO₂ reduction based on “d-d electron transfer”.

Comment 3: *Various photocatalysts of CNON, CPON and CCON were investigated, while the performance feedbacks were different, such as the activity, selectivity and the ratio of CO vs CH₄. The authors should explain the origins of these differences.*

Response: We thank the Reviewer's for the thoughtful comments. Different catalysts have different catalytic activities that are influenced by many factors. For example, they have distinct light absorption properties, and their thickness, coordination configuration, composition, surface area, and charge carrier separation abilities may also vary. Accurately defining the sources of differences in catalytic activity is a huge research challenge. In our work, we focused on studying new d-d electron transition modes for IR light-driven CO₂ reduction and demonstrated their universality. For exploring the catalytic activity sources of different catalysts, it is an interesting and challenging topic, and we will continue to work on it in the future.

Comment 4: *Line 327 in the main text, there is no definition of the symbols CNON, CPON and CCON samples, which should be added.*

Response: Thanks for the Reviewer's kind reminder. Actually, we **have identified** the symbols CNON, CPON and CCON samples at the end of "Introduction" part, please see the Line 92 in the original manuscript.

Comment 5: *In scheme 1a, the position of CB symbol is not clear. In Supplementary figure S23, the figure (c) is missed.*

Response: In scheme 1a, it represents a possible electron transfer mechanism for metallic catalysts, in which the energy bands below the Fermi level with fully occupied electrons are regarded as Conduction Band (CB). To make it clearer, we **have added** more detailed description in the revised manuscript as follows:

"(a) Metallic catalysts (the energy bands below the Fermi level with fully occupied electrons are regarded as CB and the first band above the Fermi level without electron occupation is called to B₁)"

For Supplementary Figure 23 (in the original manuscript), we **have added** the corresponding figure c icon in the revised Supporting Information. We are sorry for our carelessness.

Comment 6: *In figure 3d, the VB from the UPS is 3.09 eV, it seems not the same position with that showed in figure 3f. Why?*

Response: Thanks for the Reviewer's concerns. The SRPES valence-band spectra were measured using synchrotron-radiation light as the excitation source with a photon energy of 40.00 eV (performed at the Catalysis and Surface Science Endstation at the BL11U beamline of the National Synchrotron Radiation Laboratory), while the value of the valence band maxima (VBM) shown in Fig. 3d and Supplementary Figure 13a are referenced to the Fermi level determined from Au, called as E_{edge} shown in Fig. N14. The valence band maxima (VBM) of the samples referenced to Normal Hydrogen Electrode (NHE) can be obtained according to the following equations (*Angew. Chem. Int. Ed.* 2019, 58, 11791-11795; *Nat. Energy* 2021, 6, 388-397):

$$\Phi = h\nu - E_{\text{cutoff}}$$

$$E_{\text{VBM}} = E_{\text{edge}} + \Phi - 4.5 \quad (\text{vs NHE, pH} = 0)$$

where Φ is the work function, $h\nu$ is the photon energy of the excitation source, E_{cutoff} is the energy of secondary electron cutoff, and E_{VBM} is the valence band maxima of samples vs NHE at $\text{pH} = 0$. According to the above, we can get the E_{VBM} of c-CSON (2.42 V in Fig. 3f) and p-CSON (2.70 V in Supplementary Figure 13c).

Fig. N14 (Supplementary Figure 14) | Schematic illustration of the energy level information from UPS.

To make it clearer, we have added the above description to the revised manuscript and modified the corresponding expression.

REVIEWERS' COMMENTS

Reviewer #1 (Remarks to the Author):

Accepted as it is.

Reviewer #2 (Remarks to the Author):

I am satisfied with the revisions that the authors have made. The manuscript can be accepted for publication as it is.

Reviewer #3 (Remarks to the Author):

The authors have addressed all the issues, therefore I recommend publication without further changes.

Reviewer #4 (Remarks to the Author):

The revised version can be accepted.

Responses to the Reviewers' comments to the manuscript: NCOMMS-23-08584B. We would like to thank all the reviewers for the insightful comments and suggestions, and for their time in helping us to improve this manuscript.

Point-to-point Response to Reviewer #1

Overall comments: *Accepted as it is.*

Overall response: We greatly appreciate the reviewer for agreeing to recommend our article for publication.

Point-to-point Response to Reviewer #2

Overall comments: *I am satisfied with the revisions that the authors have made. The manuscript can be accepted for publication as it is.*

Overall response: We thank the Reviewer for agreeing to recommend our article for publication.

Point-to-point Response to Reviewer #3

Overall comments: *The authors have addressed all the issues, therefore I recommend publication without further changes.*

Overall response: We thank the Reviewer for agreeing to recommend our article for publication.

Point-to-point Response to Reviewer #4

Overall comments: The revised version can be accepted.

Overall response: We thank the Reviewer for agreeing to recommend our article for publication.